# Entropy-dissipation Informed Neural Network for McKean-Vlasov Type PDEs

**Zebang Shen**[*]
ETH Zürich
zebang.shen@inf.ethz.ch

**Zhenfu Wang**[*]
Peking University
zwang@bicmr.pku.edu.cn

## Abstract

The McKean-Vlasov equation (MVE) describes the collective behavior of particles subject to drift, diffusion, and mean-field interaction. In physical systems, the interaction term can be singular, i.e. it diverges when two particles collide. Notable examples of such interactions include the Coulomb interaction, fundamental in plasma physics, and the Biot-Savart interaction, present in the vorticity formulation of the 2D Navier-Stokes equation (NSE) in fluid dynamics. Solving MVEs that involve singular interaction kernels presents a significant challenge, especially when aiming to provide rigorous theoretical guarantees. In this work, we propose a novel approach based on the concept of entropy dissipation in the underlying system. We derive a potential function that effectively controls the KL divergence between a hypothesis solution and the ground truth. Building upon this theoretical foundation, we introduce the Entropy-dissipation Informed Neural Network (`EINN`) framework for solving MVEs. In `EINN`, we utilize neural networks (NN) to approximate the underlying velocity field and minimize the proposed potential function. By leveraging the expressive power of NNs, our approach offers a promising avenue for tackling the complexities associated with singular interactions. To assess the empirical performance of our method, we compare `EINN` with SOTA NN-based MVE solvers. The results demonstrate the effectiveness of our approach in solving MVEs across various example problems.

## 1 Introduction

Scientists use Partial Differential Equations (PDEs) to describe natural laws and predict the dynamics of real-world systems. As PDEs are of fundamental importance, a growing area in machine learning is the use of neural networks (NN) to solve these equations [Han et al., 2018, Zhang et al., 2018, Raissi et al., 2020, Cai et al., 2021, Karniadakis et al., 2021, Cuomo et al., 2022]. An important category of PDEs is the McKean-Vlasov equation (MVE), which models the dynamics of a stochastic particle system with mean-field interactions

$$d\mathbf{X}_t = -\nabla V(\mathbf{X}_t)dt + K * \bar{\rho}_t(\mathbf{X}_t)dt + \sqrt{2\nu}d\mathbf{B}_t, \quad \bar{\rho}_t = \text{Law}(\mathbf{X}_t). \tag{1}$$

Here $\mathbf{X}_t \in \mathcal{X}$ denotes a random particle' position, $\mathcal{X}$ is either $\mathbb{R}^d$ or the torus $\Pi^d$ (a cube $[-L, L]^d$ with periodic boundary condition), $V : \mathbb{R}^d \to \mathbb{R}$ denotes a *known* potential, $K : \mathbb{R}^d \to \mathbb{R}^d$ denotes some interaction kernel and the convolution operation is defined as $h * \phi = \int_{\mathcal{X}} h(\boldsymbol{x} - \boldsymbol{y})\phi(\boldsymbol{y})d\boldsymbol{y}$, $\{\mathbf{B}_t\}_{t \geq 0}$ is the standard $d$-dimensional Wiener process with $\nu \geq 0$ being the diffusion coefficient, and $\bar{\rho}_t : \mathcal{X} \to \mathbb{R}$ is the law or the probability density function of the random variable $\mathbf{X}_t$ and the initial data $\bar{\rho}_0$ is given. Under mild regularity conditions, the density function $\bar{\rho}_t$ satisfies the MVE

$$\text{(MVE)} \quad \partial_t \bar{\rho}_t(\boldsymbol{x}) + \text{div}\left(\bar{\rho}_t(-\nabla V(\boldsymbol{x}) + K * \bar{\rho}_t(\boldsymbol{x}))\right) = \nu \Delta \bar{\rho}_t(\boldsymbol{x}), \tag{2}$$

---

[*]Authors are listed in alphabetic order.

37th Conference on Neural Information Processing Systems (NeurIPS 2023).

where $\mathrm{div}$ denotes the divergence operator, $\mathrm{div}\, h(\boldsymbol{x}) = \sum_{i=1}^d \partial h_i / \partial x_i$ for a velocity field $h : \mathbb{R}^d \to \mathbb{R}^d$, $\Delta$ denotes the Laplacian operator defined as $\Delta \phi = \mathrm{div}(\nabla \phi)$, where $\nabla \phi$ denotes the gradient of a scalar function $\phi : \mathbb{R}^d \to \mathbb{R}$. Note that all these operators are applied only on the spatial variable $\boldsymbol{x}$.

In order to describe dynamics in real-world phenomena such as electromagnetism [Golse, 2016] and fluid mechanics [Majda et al., 2002], the interaction kernels $K$ in the MVE can be highly *singular*, i.e. $\|K(\boldsymbol{x})\| \to \infty$ when $\|\boldsymbol{x}\| \to 0$. Two of the most notable examples are the Coulomb interactions

$$\text{(Coulomb Kernel)} \quad K(\boldsymbol{x}) = -\nabla g(\boldsymbol{x}), \text{ with } g(\boldsymbol{x}) = \begin{cases} ((d-2)S_{d-1}(1))^{-1} \|\boldsymbol{x}\|^{-(d-2)}, & d \geq 3, \\ -(2\pi)^{-1} \log \|\boldsymbol{x}\|, & d = 2, \end{cases} \quad (3)$$

with $S_{d-1}(1)$ denoting the surface area of the unit sphere in $\mathbb{R}^d$, and the vorticity formulation of the 2D Navier-Stokes equation (NSE) where the interaction kernel $K$ is given by the Biot-Savart law

$$\text{(Biot-Savart Kernel)} \quad K(\boldsymbol{x}) = \frac{1}{2\pi} \frac{\boldsymbol{x}^\perp}{\|\boldsymbol{x}\|^2} = \frac{1}{2\pi} \left( -\frac{x_2}{\|\boldsymbol{x}\|^2}, \frac{x_1}{\|\boldsymbol{x}\|^2} \right), \quad (4)$$

where $\boldsymbol{x} = (x_1, x_2)$ and $\|\boldsymbol{x}\|$ denotes the Euclidean norm of a vector.

Classical methods for solving MVEs, including finite difference, finite volume, finite element, spectral methods, and particle methods, have been developed over time. A common drawback of these methods lies in the constraints of their solution representation: Sparse representations, such as less granular grids, cells, meshes, fewer basis functions, or particles, may lead to an inferior solution accuracy; On the other hand, dense representations incur higher computational and memory costs.

As a potent tool for function approximation, NNs are anticipated to overcome these hurdles and handle higher-dimensional, less regular, and more complex systems efficiently [Weinan et al., 2021]. The most renowned NN-based algorithm is the Physics Informed Neural Network (PINN) [Raissi et al., 2019]. The philosophy behind the PINN method is that solving a PDE system is equivalent to finding the root of the corresponding differential operators. PINN tackles the latter problem by directly parameterizing the hypothesis solution with an NN and training it to minimize the $\mathcal{L}^2$ functional residual of the operators. As a versatile PDE solver, PINN may fail to exploit the underlying dynamics of the PDE, which possibly leads to inferior performance on task-specific solvers. For example, on the 2D NSE problem, a recent NN-based development Zhang et al. [2022] surpasses PINN and sets a new SOTA empirical performance, which however lacks rigorous theoretical substantiation. Despite the widespread applications of PINN, rigorous error estimation guarantees are scarce in the literature. While we could not find results on the MVE with the Coulomb interaction, only in a very recent paper [De Ryck et al., 2023], the authors establish for NSE that the PINN loss controls the discrepancy between a candidate solution and the ground truth. We highlight that their result holds *average-in-time*, meaning that at a particular timestamp $t \in [0, T]$, a candidate solution with small PINN loss may still significantly differ from the true solution. In contrast, all guarantees in this paper are *uniform-in-time*. Moreover, there is a factor in the aforementioned guarantee that *exponentially depends* on the total evolving time $T$, while the factor in our guarantee for the NSE is *independent* of $T$. We highlight that these novel improvements are achieved for the proposed EINN framework since we take a completely different route from PINN: Our approach is explicitly designed to exploit the underlying dynamics of the system, as elaborated below.

**Our approach** Define the operator

$$\mathcal{A}[\rho] \stackrel{\text{def}}{=} -\nabla V + K * \rho - \nu \nabla \log \rho. \quad (5)$$

By noting $\Delta \bar{\rho}_t = \mathrm{div}(\bar{\rho}_t \nabla \log \bar{\rho}_t)$, we can rewrite the MVE in the form of a continuity equation

$$\partial_t \bar{\rho}_t(\boldsymbol{x}) + \mathrm{div}\Big( \bar{\rho}_t(\boldsymbol{x}) \mathcal{A}[\bar{\rho}_t](\boldsymbol{x}) \Big) = 0. \quad (6)$$

For simplicity, we will refer to $\mathcal{A}[\bar{\rho}_t]$ as the *underlying velocity*. Consider another time-varying *hypothesis velocity* field $f : \mathbb{R} \times \mathbb{R}^d \to \mathbb{R}$ and let $\rho_t^f$ be the solution to the continuity equation

$$\text{(hypothesis solution)} \quad \partial_t \rho_t^f(\boldsymbol{x}) + \mathrm{div}(\rho_t^f(\boldsymbol{x}) f(t, \boldsymbol{x})) = 0, \ \rho_0^f = \bar{\rho}_0 \quad (7)$$

for $t \in [0, T]$, where we recall that the initial law $\bar{\rho}_0$ is known. We will refer to $\rho_t^f$ as the *hypothesis solution* and use the superscript to emphasize its dependence on the hypothesis velocity field $f$. We propose an Entropy-dissipation Informed Neural Network framework (EINN), which trains an NN parameterized hypothesis velocity field $f_\theta$ by minimizing the following EINN loss

$$\text{(EINN loss)} \quad R(f_\theta) \stackrel{\text{def}}{=} \int_0^T \int_{\mathcal{X}} \|f_\theta(t, \boldsymbol{x}) - \mathcal{A}[\rho_t^{f_\theta}](\boldsymbol{x})\|^2 \rho_t^{f_\theta}(\boldsymbol{x}) \mathrm{d}\boldsymbol{x} \mathrm{d}t. \quad (8)$$

The objective (8) is obtained by studying the stability of carefully constructed Lyapunov functions. These Lyapunov functions draw inspiration from the concept of entropy dissipation in the system, leading to the name of our framework. We highlight that we provide a rigorous error estimation guarantee for our framework for MVEs with singular kernels (3) and (4), showing that when $R(f_\theta)$ is sufficiently small, $\rho_t^{f_\theta}$ recovers the ground truth $\bar{\rho}_t$ in the KL sense, uniform in time.

**Theorem 1** (Informal). *Suppose that the initial density function $\bar{\rho}_0$ is sufficiently regular and the hypothesis velocity field $f_t(\cdot) = f(t, \cdot)$ is at least three times continuously differentiable both in $t$ and $x$. We have for the MVE with a bounded interaction kernel $K$ or with the singular Coulomb (3) or Biot-Savart (4) interaction, the KL divergence between the hypothesis solution $\rho_t^f$ and the ground truth $\bar{\rho}_t$ is controlled by the* `EINN` *loss for any $t \in [0, T]$, i.e. there exists some constant $C > 0$,*

$$\sup_{t \in [0,T]} \mathbf{KL}(\rho_t^f, \bar{\rho}_t) \le CR(f). \tag{9}$$

Having stated our main result, we elaborate on the difference between `EINN` and `PINN` in terms of information flow over time, which explains why `EINN` achieves better theoretical guarantees: In `PINN`, the residuals at different time stamps are independent of each other and hence there is no information flow from the residual at time $t_1$ to the one at time $t_2(> t_1)$. In contrast, in the `EINN` loss (8), incorrect estimation made in $t_1$ will also affect the error at $t_2$ through the hypothesis solution $\rho_t^f$. Such an information flow gives a stronger gradient signal when we are trying to minimize the `EINN` loss, compared to the `PINN` loss. It partially explains why we can obtain the novel uniform-in-time estimation as opposed to the average-in-time estimation for `PINN` and why the constant $C$ in the NSE case is independent of $T$ for `EINN` (Theorem 2), but exponential in $T$ for `PINN`.

**Contributions.** In summary, we present a novel NN-based framework for solving the MVEs. Our method capitalizes on the entropy dissipation property of the underlying system, ensuring robust theoretical guarantees even when dealing with singular interaction kernels. We elaborate on the contributions of our work from theory, algorithm, and empirical perspectives as follows.

1. (Theory-wise) By studying the stability of the MVEs with bounded interaction kernels or with singular interaction kernels in the Coulomb (3) and the Biot-Savart case (4) (the 2D NSE) via entropy dissipation, we establish the error estimation guarantee for the `EINN` loss on these equations. Specifically, we design a potential function $R(f)$ of a hypothesis velocity $f$ such that $R(f)$ controls the KL divergence between the hypothesis solution $\rho_t^f$ (defined in equation (7)) and the ground truth solution $\bar{\rho}_t$ for any time stamp within a given time interval $[0, T]$. A direct consequence of this result is that $R(f)$ can be used to assess the quality of a generic hypothesis solution to the above MVEs and $\rho_t^f$ exactly recovers $\bar{\rho}_t$ in the KL sense given that $R(f) = 0$.

2. (Algorithm-wise) When the hypothesis velocity field is parameterized by an NN, i.e. $f = f_\theta$ with $\theta$ being some finite-dimensional parameters, the `EINN` loss $R(f_\theta)$ can be used as the loss function of the NN parameters $\theta$. We discuss in detail how an estimator of the gradient $\nabla_\theta R(f_\theta)$ can be computed so that stochastic gradient-based optimizers can be utilized to train the NN. In particular, for the 2D NSE (the Biot-Savart case (4)), we show that the singularity in the gradient computation can be removed by exploiting the anti-derivative of the Biot-Savart kernel.

3. (Empirical-wise) We compare the proposed approach, derived from our novel theoretical guarantees, with SOTA NN-based algorithms for solving the MVE with the Coulomb interaction and the 2D NSE (the Biot-Savart interaction). We pick specific instances of the initial density $\bar{\rho}_0$, under which explicit solutions are known and can be used as the ground truth to test the quality of the hypothesis ones. Using NNs with the same complexity (depth, width, and structure), we observe that the proposed method significantly outperforms the included baselines.

## 2 Entropy-dissipation Informed Neural Network

In this section, we present the proposed `EINN` framework for the MVE. To understand the intuition behind our design, we first write the continuity equation (7) in a similar form as the MVE (6):

$$\partial_t \rho_t^f(\boldsymbol{x}) + \mathrm{div}\left( \rho_t^f(\boldsymbol{x}) \Big( \mathcal{A}[\rho_t^f](\boldsymbol{x}) + \delta_t(\boldsymbol{x}) \Big) \right) = 0, \tag{10}$$

where $f$ is the hypothesis velocity (recall that $f_t(\cdot) = f(t, \cdot)$) and

$$\text{(Perturbation)} \quad \delta_t(\boldsymbol{x}) \overset{\text{def}}{=} f_t(\boldsymbol{x}) - \mathcal{A}[\rho_t^f](\boldsymbol{x}) \tag{11}$$

can be regarded as a perturbation to the original MVE system. Consequently, it is natural to study the deviation of the hypothesis solution $\rho_t^f$ from the true solution $\bar{\rho}_t$ using an appropriate Lyapunov

function $L(\rho_t^f, \bar{\rho}_t)$. The functional relation between this deviation and the perturbation is termed as the *stability* of the underlying dynamical system, which allows us to derive the `EINN` loss (8). Following this idea, the design of the `EINN` loss can be determined by the choice of the Lyapunov function $L$ used in the stability analysis. In the following, we describe the Lyapunov function used for the MVE with the Coulomb interaction and the 2D NSE (MVE with Biot-Savart interaction). The proof of the following results are the major theoretical contributions of this paper and will be elaborated in the analysis section 3.

- For the MVE with the Coulomb interaction, we choose $L$ to be the *modulated free energy $E$* (defined in equation (23)) which is originally proposed in [Bresch et al., 2019a] to establish the mean-field limit of a corresponding interacting particle system. We have (setting $L = E$)

$$\frac{\mathrm{d}}{\mathrm{d}t} E(\rho_t^f, \bar{\rho}_t) \le \frac{1}{2} \int_{\mathcal{X}} \rho_t^f(\boldsymbol{x}) \|\delta_t(\boldsymbol{x})\|^2 \mathrm{d}\boldsymbol{x} + C\, E(\rho_t^f, \bar{\rho}_t), \tag{12}$$

  where $C$ is a universal constant depending on $\nu$ and $(\bar{\rho}_t)_{t \in [0,T]}$.

- For the 2D NSE (MVE with the Biot-Savart interaction), we choose $L$ as the KL divergence. Our analysis is inspired by [Jabin and Wang, 2018] which for the first time establishes the quantitative mean-field limit of the stochastic interacting particle systems where the interaction kernel can be in some negative Sobolev space. We have

$$\frac{\mathrm{d}}{\mathrm{d}t} \mathbf{KL}(\rho_t^f, \bar{\rho}_t) \le -\frac{\nu}{2} \int_{\mathcal{X}} \rho_t^f(\boldsymbol{x}) \|\nabla \log \frac{\rho_t^f}{\bar{\rho}_t}(\boldsymbol{x})\|^2 + C\,\mathbf{KL}(\rho_t^f, \bar{\rho}_t) + \frac{1}{\nu} \int_{\mathcal{X}} \rho_t^f(\boldsymbol{x}) \|\delta_t(\boldsymbol{x})\|^2 \mathrm{d}\boldsymbol{x}, \tag{13}$$

  where again $C$ is a universal constant depending on $\nu$ and $(\bar{\rho}_t)_{t \in [0,T]}$.

After applying Grönwall's inequality on the above results, we can see that the `EINN` loss (8) is precisely the term derived by stability analysis of the MVE system with an appropriate Lyapunov function. In the next section, we elaborate on how a stochastic approximation of $\nabla_\theta R(f_\theta)$ can be efficiently computed for a parameterized hypothesis velocity field $f = f_\theta$ so that stochastic optimization methods can be utilized to minimize $R(f_\theta)$.

## 2.1 Stochastic Gradient Computation with Neural Network Parameterization

While the choice of the `EINN` loss (8) is theoretically justified through the above stability study, in this section, we show that it admits an estimator which can be efficiently computed. Define the flow map $X_t$ via the ODE $\mathrm{d}\boldsymbol{x}(t) = f_t(\boldsymbol{x}(t); \theta)\mathrm{d}t$ with $\boldsymbol{x}(0) = \boldsymbol{x}_0$ such that $\boldsymbol{x}(t) = X_t(\boldsymbol{x}_0)$. From the definition of the push-forward measure, one has $\rho_t^f = X_t \sharp \bar{\rho}_0$. Recall the definitions of the `EINN` loss $R(f)$ in equation (8) and the perturbation $\delta_t$ in equation (11). Use the change of variable formula of the push-forward measure in (a) and Fubini's theorem in (b). We have

$$R(f) = \int_0^T \|\delta_t\|_{\rho_t^f}^2 dt \overset{(a)}{=} \int_0^T \|\delta_t \circ X_t\|_{\bar{\rho}_0}^2 dt \overset{(b)}{=} \int \int_0^T \|\delta_t \circ X_t(\boldsymbol{x}_0)\|^2 dt\, d\bar{\rho}_0(\boldsymbol{x}_0). \tag{14}$$

Consequently, by defining the trajectory-wise loss (recall $\boldsymbol{x}(t) = X_t(\boldsymbol{x}_0)$)

$$R(f; \boldsymbol{x}_0) = \int_0^T \|\delta_t \circ X_t(\boldsymbol{x}_0)\|^2 dt = \int_0^T \|\delta_t(\boldsymbol{x}(t))\|^2 dt, \tag{15}$$

we can write the potential function (8) as an expectation $R(f) = \mathbb{E}_{\boldsymbol{x}_0 \sim \bar{\rho}_0}[R(f; \boldsymbol{x}_0)]$. Similarly, when $f$ is parameterized as $f = f_\theta$, we obtain the expectation form $\nabla_\theta R(f_\theta) = \mathbb{E}_{\boldsymbol{x}_0 \sim \bar{\rho}_0}[\nabla_\theta R(f_\theta; \boldsymbol{x}_0)]$.

We show $\nabla_\theta R(f_\theta; \boldsymbol{x}_0)$ can be computed accurately, via the adjoint method (for completeness see the derivation of the adjoint method in appendix D). As a recap, suppose that we can write $R(f_\theta; \boldsymbol{x}_0)$ in a standard ODE-constrained form $R(f_\theta; \boldsymbol{x}_0) = \ell(\theta) = \int_0^T g(t, \boldsymbol{s}(t), \theta)dt$, where $\{\boldsymbol{s}(t)\}_{t \in [0,T]}$ is the solution to the ODE $\frac{d}{dt}\boldsymbol{s}(t) = \psi(t, \boldsymbol{s}(t); \theta)$ with $\boldsymbol{s}(0) = \boldsymbol{s}_0$, and $\psi$ is a known transition function. The adjoint method states that the gradient $\frac{d}{d\theta}\ell(\theta)$ can be computed as

$$\text{(Adjoint Method)} \quad \frac{d\ell}{d\theta} = \int_0^T a(t)^\top \frac{\partial \psi}{\partial \theta}(t, \boldsymbol{s}(t); \theta) + \frac{\partial g}{\partial \theta}(t, \boldsymbol{s}(t); \theta)\mathrm{d}t. \tag{16}$$

where $a(t)$ solves the final value problems $\frac{d}{dt}a(t)^\top + a(t)^\top \frac{\partial \psi}{\partial s}(t, \boldsymbol{s}(t); \theta) + \frac{\partial g}{\partial s}(t, \boldsymbol{s}(t); \theta) = 0, a(T) = 0$. In the following, we focus on how $R(f_\theta; \boldsymbol{x}_0)$ can be written in the above ODE-constrained form.

**Write $R(f_\theta; \boldsymbol{x}_0)$ in ODE-constrained form** Expanding the definition of $\delta_t$ in equation (11) gives

$$\delta_t(\boldsymbol{x}(t)) = f_t(\boldsymbol{x}(t)) - \left(-\nabla V(\boldsymbol{x}(t)) + K * \rho_t^f(\boldsymbol{x}(t)) - \nu\nabla\log\rho_t^f(\boldsymbol{x}(t))\right). \tag{17}$$

Note that in the above quantity, $f$ and $V$ are known functions. Moreover, it is known that $\nabla\log\rho_t^f(\boldsymbol{x}(t))$ admits a closed form dynamics (e.g. see Proposition 2 in [Shen et al., 2022])

$$\frac{d}{dt}\nabla\log\rho_t^f(\boldsymbol{x}(t)) = -\nabla\left(\operatorname{div} f_t(\boldsymbol{x}(t); \theta)\right) - \left(\mathcal{J}_{f_t}(\boldsymbol{x}(t); \theta)\right)^\top \nabla\log\rho_t^f(\boldsymbol{x}(t)), \tag{18}$$

which allows it to be explicitly computed by starting from $\nabla\log\bar\rho_0(x_0)$ and integrating over time (recall that $\bar\rho_0$ is known). Here $\mathcal{J}_{f_t}$ denotes the Jacobian matrix of $f_t$. Consequently, all we need to handle is the convolution term $K * \rho_t^f(\boldsymbol{x}(t))$.

A common choice to approximate the convolution operation is via Monte-Carlo integration: Let $\boldsymbol{y}_i(t) \overset{\text{iid}}{\sim} \rho_t^f$ for $i = 1, \ldots, N$ and denote an empirical approximation of $\rho_t^f$ by $\mu_N^{\rho_t^f} = \frac{1}{N}\sum_{i=1}^N \delta_{\boldsymbol{y}_i(t)}$, where $\delta_{\boldsymbol{y}_i(t)}$ denotes the Dirac measure at $\boldsymbol{y}_i(t)$. We approximate the convolution term in equation (17) in different ways for the Coulomb and the Biot-Savart interactions:

1. For the Coulomb type kernel (3), we first approximate $K * \rho_t^f$ with $K_c * \rho_t^f$, where

$$K_c(\boldsymbol{x}) \overset{\text{def}}{=} \begin{cases} K(\boldsymbol{x}) & \text{if } \|\boldsymbol{x}\| > c, \\ 0 & \text{if } \|\boldsymbol{x}\| \le c. \end{cases} \tag{19}$$

If $\rho_t^{\boldsymbol{f}}$ is bounded in $\mathcal{X}$, we have

$$\sup_{\boldsymbol{x}\in\mathcal{X}}\|(K - K_c) * \rho_t^f(\boldsymbol{x})\| = \sup_{\boldsymbol{x}\in\mathcal{X}}\left\|\int_{\|\boldsymbol{x}-\boldsymbol{y}\|\le c}\frac{\boldsymbol{x}-\boldsymbol{y}}{\|\boldsymbol{x}-\boldsymbol{y}\|^d}\rho_t^f(\boldsymbol{y})\mathrm{d}\boldsymbol{y}\right\| \le \|\rho_t^f\|_{\mathcal{L}^\infty(\mathcal{X})}\int_{\|\boldsymbol{y}\|\le c}\frac{1}{\|\boldsymbol{y}\|^{d-1}}\mathrm{d}\boldsymbol{y}.$$

To compute the integral on the right-hand side, we will switch to polar coordinates $(r, \psi)$:

$$\int_{|\boldsymbol{y}|\le c}\frac{1}{|\boldsymbol{y}|^{d-1}}\mathrm{d}\boldsymbol{y} = \int_0^c \mathrm{d}r\frac{1}{r^{d-1}}\int_\Psi \mathrm{d}\psi\, J_{(r,\psi)} \le \int_0^c \mathrm{d}r = c. \tag{20}$$

Here, $J_{(r,\psi)}$ denotes the determinant of the Jacobian matrix resulting from the transformation from the Cartesian system to the polar coordinate system. In inequality (20), we utilize the fact that $J_{(r,\psi)} \le r^{d-1}$, which allows us to cancel out the factor $1/r^{d-1}$. Now that $K_c$ is bounded by $c^{-d+1}$, we can further approximate $K_c * \rho_t^{\boldsymbol{f}}$ using $K_c * \mu_N^{\rho_t^{\boldsymbol{f}}}$ with error of the order $O(c^{-d+1}/\sqrt{N})$. Altogether, we have $\sup_{\boldsymbol{x}\in\mathcal{X}}\|K * \rho_t^f(\boldsymbol{x}) - K_c * \mu_N^{\rho_t^{\boldsymbol{f}}}(\boldsymbol{x})\| = O(c + c^{-d+1}/\sqrt{N})$ which can be made arbitrarily small for a sufficiently small $c$ and a sufficiently large $N$.

2. For Biot-Savart interaction (2D Navier-Stokes equation), there are more structures to exploit and we can completely avoid the singularity: As noted by Jabin and Wang [2018], the convolution kernel $K$ can be written in a divergence form:

$$K = \nabla \cdot U, \quad \text{with } U(\boldsymbol{x}) = \frac{1}{2\pi}\begin{bmatrix} -\arctan(\frac{\boldsymbol{x}_1}{\boldsymbol{x}_2}), & 0 \\ 0, & \arctan(\frac{\boldsymbol{x}_2}{\boldsymbol{x}_1}) \end{bmatrix}, \tag{21}$$

where the divergence of a matrix function is applied row-wise, i.e. $[K(\boldsymbol{x})]_i = \operatorname{div} U_i(\boldsymbol{x})$. Using integration by parts and noticing that the boundary integration vanishes on the torus, one has

$$K * \rho_t^f(\boldsymbol{x}) = \int K(\boldsymbol{y})\rho_t^f(\boldsymbol{x}-\boldsymbol{y})d\boldsymbol{y} = \int \nabla\cdot U(\boldsymbol{y})\rho_t^f(\boldsymbol{x}-\boldsymbol{y})d\boldsymbol{y} = \int U(\boldsymbol{y})\nabla\rho_t^f(\boldsymbol{x}-\boldsymbol{y})d\boldsymbol{y}$$

$$= \int U(\boldsymbol{x}-\boldsymbol{y})\rho_t^f(\boldsymbol{y})\nabla\log\rho_t^f(\boldsymbol{y})d\boldsymbol{y} = \mathbb{E}_{\boldsymbol{y}\sim\rho_t^f(\boldsymbol{y})}[U(\boldsymbol{x}-\boldsymbol{y})\nabla\log\rho_t^f(\boldsymbol{y})].$$

If the score function $\nabla\log\rho_t^f$ is bounded, then the integrand in the expectation is also bounded. Therefore, we can avoid integrating singular functions and the Monte Carlo-type estimation $\frac{1}{N}\sum_{i=1}^N U(\boldsymbol{x}-\boldsymbol{y}_i(t))\nabla\log\rho_t^f(\boldsymbol{y}_i(t))$ is accurate for a sufficiently large value of N.

With the above discussion, we can write $R(f_\theta; \boldsymbol{x}_0)$ in an ODE-constrained form in a standard way, which due to space limitation is deferred to Appendix D.1.

**Remark 1.** *Let $\ell_N(\theta)$ be the function we obtained using the above approximation of the convolution, where N is the number of Monte-Carlo samples. The above discussion shows that $\ell_N(\theta)$ and $R(f_\theta; \boldsymbol{x}_0)$ are close in the $\mathcal{L}^\infty$ sense, which is hence sufficient when the EINN loss is used as error quantification since only function value matters. When both $\ell_N(\theta)$ and $R(f_\theta; \boldsymbol{x}_0)$ are in $C^2$, one can translate the closeness in function value to the closeness of their gradients. In our experiments, using $\nabla\ell_N(\theta)$ as an approximation of $\nabla_\theta R(f_\theta; \boldsymbol{x}_0)$ gives very good empirical performance already.*

# 3 Analysis

In this section, we focus on the torus case, i.e. $\mathcal{X} = \Pi^d$ is a box with the periodic boundary condition. This is a typical setting considered in the literature as the universal function approximation of NNs only holds over a compact set. Moreover, the boundary integral resulting from integration by parts vanishes in this setting, making it amenable for analysis purposes. For completeness, we provide a discussion on the unbounded case, i.e. $\mathcal{X} = \mathbb{R}^d$ in the Appendix G, which requires additional regularity assumptions. Given the MVE (2), if $K$ is bounded, it is sufficient to choose the Lyapunov functional $L(\rho_t^f, \bar{\rho}_t)$ as the KL divergence (please see Theorem 6 in the appendix). But for the singular Coulomb kernel, we need also to consider the modulated energy as in [Serfaty, 2020]

$$\text{(Modulated Energy)} \quad F(\rho, \bar{\rho}) \overset{\text{def}}{=} \frac{1}{2} \int_{\mathcal{X}^2} g(x-y) \mathrm{d}(\rho - \bar{\rho})(x) \mathrm{d}(\rho - \bar{\rho})(y), \tag{22}$$

where $g$ is the fundamental solution to the Laplacian equation in $\mathbb{R}^d$, i.e. $-\Delta g = \delta_0$, and the Coulomb interaction reads $K = -\nabla g$ (see its closed form expression in equation (3)). If we are only interested in the deterministic dynamics with Coulomb interactions, i.e. $\nu = 0$ in equation (2), it suffices to choose $L(\rho_t^f, \bar{\rho})$ as $F(\rho_t^f, \bar{\rho}_t)$ (please see Theorem 3). But if we consider the system with Coulomb interactions and diffusions, i.e. $\nu > 0$, we shall combine the KL divergence and the modulated energy to form the modulated free energy as in Bresch et al. [2019b], which reads

$$\text{(Modulated Free Energy)} \quad E(\rho, \bar{\rho}) \overset{\text{def}}{=} \nu \mathbf{KL}(\rho, \bar{\rho}) + F(\rho, \bar{\rho}). \tag{23}$$

This definition agrees with the physical meaning that "Free Energy = Temperature × Entropy + Energy", and we note that the temperature is proportional to the diffusion coefficient $\nu$. We remark also for two probability densities $\rho$ and $\bar{\rho}$, $F(\rho, \bar{\rho}) \geq 0$ since by looking in the Fourier domain $F(\rho, \bar{\rho}) = \int \hat{g}(\xi) |\widehat{\rho - \bar{\rho}}(\xi)|^2 \mathrm{d}\xi \geq 0$ as $\hat{g}(\xi) \geq 0$. Moreover, $F(\rho, \bar{\rho})$ can be regarded as a negative Sobolev norm for $\rho - \bar{\rho}$, which metricizes weak convergence.

To obtain our main stability estimate, we first obtain the time evolution of the KL divergence.

**Lemma 1** (Time Evolution of the KL divergence). *Given the hypothesis velocity field $f = f(t, x) \in C_{t,x}^1$. Assume that $(\rho_t^f)_{t \in [0,T]}$ and $(\bar{\rho}_t)_{t \in [0,T]}$ are classical solutions to equation (7) and equation (6) respectively. It holds that (recall the definition of $\delta_t$ in equation (11))*

$$\frac{\mathrm{d}}{\mathrm{d}t} \int_{\mathcal{X}} \rho_t^f \log \frac{\rho_t^f}{\bar{\rho}_t} = -\nu \int_{\mathcal{X}} \rho_t^f |\nabla \log \frac{\rho_t^f}{\bar{\rho}_t}|^2 + \int_{\mathcal{X}} \rho_t^f K * (\rho_t^f - \bar{\rho}_t) \cdot \nabla \log \frac{\rho_t^f}{\bar{\rho}_t} + \int_{\mathcal{X}} \rho_t^f \delta_t \cdot \nabla \log \frac{\rho_t^f}{\bar{\rho}_t},$$

*where $\mathcal{X}$ is the tours $\Pi^d$. All the integrands are evaluated at $\boldsymbol{x}$.*

We refer the proof of this lemma and all other lemmas and theorems in this section to the appendix E. We remark that to have the existence of classical solution $(\bar{\rho}_t)_{t \in [0,T]}$, we definitely need the regularity assumptions on $-\nabla V$ and on $K$. But the linear term $-\nabla V$ will not contribute to the evolution of the relative entropy. See [Jabin and Wang, 2018] for detailed discussions.

Similarly, we have the time evolution of the modulated energy as follows.

**Lemma 2** (Time evolution of the modulated energy). *Under the same assumptions as in Lemma 1, given the diffusion coefficient $\nu \geq 0$, it holds that (recall the definition of $\delta_t$ in equation (11))*

$$\frac{\mathrm{d}}{\mathrm{d}t} F(\rho_t^f, \bar{\rho}_t) = -\int_{\mathcal{X}} \rho_t^f \|K * (\rho_t^f - \bar{\rho}_t)\|^2 - \int_{\mathcal{X}} \rho_t^f \delta_t \cdot K * (\rho_t^f - \bar{\rho}_t) + \nu \int_{\mathcal{X}} \rho_t^f K * (\rho_t^f - \bar{\rho}_t) \cdot \nabla \log \frac{\rho_t^f}{\bar{\rho}_t}$$
$$- \frac{1}{2} \int_{\mathcal{X}^2} K(x - y) \cdot \Big( \mathcal{A}[\bar{\rho}_t](x) - \mathcal{A}[\bar{\rho}_t](y) \Big) \mathrm{d}(\rho_t^f - \bar{\rho}_t)^{\otimes 2}(x, y)$$

*where we recall that the operator $\mathcal{A}$ is defined in equation (5).*

By Lemma 1 and careful analysis, in particular by rewriting the Biot-Savart law in the divergence of a bounded matrix-valued function (21), we obtain the following estimate for the 2D NSE.

**Theorem 2** (Stability estimate of the 2D NSE). *Notice that when $K$ is the Biot-Savart kernel, $\mathrm{div} K = 0$. Assume that the initial data $\bar{\rho}_0 \in C^3(\Pi^d)$ and there exists $c > 1$ such that $\frac{1}{c} \leq \bar{\rho}_0 \leq c$. Assume further the hypothesis velocity field $f(t, x) \in C_{t,x}^1$. Then it holds that*

$$\sup_{t \in [0,T]} \int_{\Pi^d} \rho_t^f \log \frac{\rho_t^f}{\bar{\rho}_t} \mathrm{d}x \leq \frac{e^C}{\nu} R(f),$$

*where $C = \int_0^\infty M(t) \mathrm{d}t < \infty$ with $M(t) \overset{\text{def}}{=} \|\nabla \log \bar{\rho}_t\|_{L^\infty}^2 / 2\nu + 2 \left\| \nabla^2 \bar{\rho}_t / \bar{\rho}_t \right\|_{L^\infty}$.*

We remark that given $\bar{\rho}_0$ is smooth enough and fully supported on $\mathcal{X}$, one can propagate the regularity to finally show the finiteness of $C$. See detailed computations as in Guillin et al. [2021]. We give the complete proof in the appendix E. This theorem tells us that as long as $R(f)$ is small, the KL divergence between $\rho_t^f$ and $\bar{\rho}_t$ is small and the control is uniform in time $t \in [0, T]$ for any $T$. Moreover, we highlight that $C$ is independent of $T$, and our result on the NSE is significantly better than the average-in-time and exponential-in-$T$ results from [De Ryck et al., 2023].

To treat the MVE (2) with Coulomb interactions, we exploit the time evolution of the modulated free energy $E(\rho_t^f, \bar{\rho}_t)$. Indeed, combining Lemma 1 and Lemma 2, we arrive at the following identity.

**Lemma 3** (Time evolution of the modulated free energy). *Under the same assumptions as in Lemma 1, one has (recall the definitions of $\delta_t$ and $\mathcal{A}$ in (11) and (5) respectively)*

$$\frac{\mathrm{d}}{\mathrm{d}t} E(\rho_t^f, \bar{\rho}_t) = -\int_{\mathcal{X}} \rho_t^f \left| K * (\rho_t^f - \bar{\rho}_t) - \nu \nabla \log \frac{\rho_t^f}{\bar{\rho}_t} \right|^2 - \int_{\mathcal{X}} \rho_t^f \, \delta_t \cdot \left( K * (\rho_t^f - \bar{\rho}_t) - \nu \nabla \log \frac{\rho_t^f}{\bar{\rho}_t} \right)$$
$$- \frac{1}{2} \int_{\mathcal{X}^2} K(x - y) \cdot \left( \mathcal{A}[\bar{\rho}_t](x) - \mathcal{A}[\bar{\rho}_t](y) \right) \mathrm{d}(\rho_t^f - \bar{\rho}_t)^{\otimes 2}(x, y).$$

Inspired by the mean-field convergence results as in Serfaty [2020] and Bresch et al. [2019b], we finally can control the growth of $E(\rho_t^f, \bar{\rho}_t)$ in the case when $\nu > 0$, and $F(\rho_t^f, \bar{\rho}_t)$ in the case when $\nu = 0$. Note also that $E(\rho_t^f, \bar{\rho}_t)$ can also control the KL divergence when $\nu > 0$.

**Theorem 3** (Stability estimate of MVE with Coulomb interactions). *Assume that for $t \in [0, T]$, the underlying velocity field $\mathcal{A}[\bar{\rho}_t](x)$ is Lipschitz in $x$ and $\sup_{t \in [0,T]} \|\nabla \mathcal{A}[\bar{\rho}_t](\cdot)\|_{L^\infty} = C_1 < \infty$. Then there exists $C > 0$ such that*

$$\sup_{t \in [0,T]} \nu \, \mathbf{KL}(\rho_t^f, \bar{\rho}_t) \leq \sup_{t \in [0,T]} E(\rho_t^f, \bar{\rho}_t) \leq \exp(CC_1 T) R(f).$$

*In the deterministic case when $\nu = 0$, under the same assumptions, it holds that*

$$\sup_{t \in [0,T]} F(\rho_t^f, \bar{\rho}_t) \leq \exp(CC_1 T) R(f).$$

Recall the definition of the operator $\mathcal{A}$ in equation 5. Given that $\mathcal{X} = \Pi^d$, and $\bar{\rho}_0$ is smooth enough and bounded from below, one can propagate regularity to obtain the Lipschitz condition for $\mathcal{A}[\bar{\rho}_t]$. See the proof and the discussion on the Lipschitz assumptions on $\mathcal{A}[\bar{\rho}_t](\cdot)$ in the appendix E.

**Approximation Error of Neural Network** Theorems 2 and 3 provide the error estimation guarantee for the proposed `EINN` loss (8). Suppose that we parameterize the velocity field $f = f_\theta$ with an NN parameterized by $\theta$, as we did in Section 2.1 and let $\tilde{f}$ be the output of an optimization procedure when $R(f_\theta)$ is used as objective. In order the explicitly quantify the mismatch between $\rho_t^{\tilde{f}}$ and $\bar{\rho}_t$, we need to quantify two errors: (i) Approximation error, reflecting how well the ground truth solution can be approximated among the NN function class of choice; (ii) Optimization error, involving minimization of a highly nonlinear non-convex objective. In the following, we show that for a function class $\mathcal{F}$ with sufficient capacity, there exists at least one element $\hat{f} \in \mathcal{F}$ that can reduce the loss function $R(\hat{f})$ as much as desired. We will not discuss how to identify such an element in the function class $\mathcal{F}$ as it is independent of our research and remains possibly the largest open problem in modern AI research. To establish our result, we make the following assumptions.

**Assumption 1.** *$\rho_0$ is sufficiently regular, such that $\nabla \log \rho_0 \in \mathcal{L}^\infty(\mathcal{X})$ and $\bar{f}_t = \mathcal{A}[\bar{\rho}_t] \in W^{2,\infty}(\mathcal{X})$. $\nabla V$ is Lipschitz continuous. Here $W^{2,\infty}(\mathcal{X})$ stands for the Sobolev norm of order $(2, \infty)$ over $\mathcal{X}$.*

We here again need to propagate the regularity for $f_t$ at least for a time interval $[0, T]$. It is easy to do so for the torus case, but for the unbounded domain, there are some technical issues to be overcome. Similar assumptions are also needed in some mathematical works for instance in Jabin and Wang [2018]. We also make the following assumption on the capacity of the function class $\mathcal{F}$, which is satisfied for example by NNs with tanh activation function [De Ryck et al., 2021].

**Assumption 2.** *The function class is sufficiently large, such that there exists $\hat{f} \in \mathcal{F}$ satisfying $\hat{f}_t \in \mathcal{C}^3(\mathcal{X})$ and $\|\hat{f}_t - \bar{f}_t\|_{W^{2,\infty}(\mathcal{X})} \leq \epsilon$ for all $t \in [0, T]$.*

**Theorem 4.** *Consider the case where the domain is the torus. Suppose that Assumptions 1 and 2 hold. For both the Coulomb and the Biot-Savart cases, there exists $\hat{f} \in \mathcal{F}$ such that $R(\hat{f}) \leq C(T) \cdot (\epsilon \cdot \ln 1/\epsilon)^2$, where $C(T)$ is some constant independent of $\epsilon$. Here $R$ is the `EINN` loss (8).*

The major difficulty to overcome is the lack of Lipschitz continuity due to the singular interaction. We successfully address this challenge by establishing that the contribution of the singular region $(\|\boldsymbol{x}\| \le \epsilon)$ to $R(\hat{f})$ can be bounded by $O((\epsilon \log \frac{1}{\epsilon})^2)$. Please see the detailed proof in Appendix F.

## 4 Related Works on NN-based PDE solvers

Solving PDEs is a key aspect of scientific research, with a wealth of literature [Evans, 2022]. Due to space limitations, a detailed discussion about the classical PDE solvers is deferred to Appendix A. In this section, we focus on the NN-based approaches as they are more related to our research.

As previously mentioned, PINN is possibly the most well-known method of this type. PINN regards the solution to a PDE system as the root of the corresponding operators $\{\mathcal{D}_i(\boldsymbol{g})\}_{i=1}^n$, and expresses the time and space boundary conditions as $\mathcal{B}(\boldsymbol{g}) = 0$, where $\boldsymbol{g}$ is a candidate solution and $\mathcal{D}_i$ and $\mathcal{B}$ are operators acting on $\boldsymbol{g}$. Parameterizing $\boldsymbol{g} = \boldsymbol{g}_\theta$ using an NN, PINN optimizes its parameters $\theta$ by minimizing the residual $L(\theta) \overset{\text{def}}{=} \sum_{i=1}^n \lambda_i \|\mathcal{D}_i(\boldsymbol{g}_\theta)\|_{\mathcal{L}^2(\mathcal{X})}^2 + \lambda_0 \|\mathcal{B}(\boldsymbol{g}_\theta)\|_{\mathcal{L}^2(\mathcal{X})}^2$. The hyperparameters $\lambda_i$ balance the validity of PDEs and boundary conditions under consideration and must be adjusted for optimal performance. In contrast, EINN requires no hyperparameter tuning. PINN is versatile and can be applied to a wide range of PDEs, but its performance may not be as good as other NN-based solvers tailored for a particular class of PDEs, as it does not take into account other in-depth properties of the system, a phenomenon observed in the literature [Krishnapriyan et al., 2021, Wang et al., 2022]. [Shin and Em Karniadakis, 2020] initiates the work of theoretically establishing the consistency of PINN by considering the linear elliptic and parabolic PDEs, for which they prove that a vanishing PINN loss $L(\theta)$ asymptotically implies $\boldsymbol{g}_\theta$ recovers the true solution. A similar result is extended to the linear advection equations in [Shin et al., 2020]. Leveraging the stability of the operators $\mathcal{D}_i$ (corresponding to PDEs of interest), non-asymptotic error estimations are established for linear Kolmogorov equations in [De Ryck and Mishra, 2022], for semi-linear and quasi-linear parabolic equations and the incompressible Euler in [Mishra and Molinaro, 2022], and for the NSE in [De Ryck et al., 2023]. We highlight these non-asymptotic results are all average-in-time, meaning that even when the PINN loss is small the deviation of the candidate solution to the true solution may be significant at a particular timestamp $t \in [0, T]$. In comparison, our results are uniform-in-time, i.e. the supremum of the deviation is strictly bounded by the EINN loss. Moreover, we show in Theorem 2, for the NSE our error estimation holds for any $T$ uniformly, while the results in [De Ryck et al., 2023] have an exponential dependence on $T$.

Recent work from Zhang et al. [2022] proposes the Random Deep Vortex Network (RDVN) method for solving the 2D NSE and achieves SOTA performance for this task. Let $\boldsymbol{u}_t^\theta$ be an estimation of the interaction term $K * \rho_t$ in the SDE (1) and use $\rho_t^\theta$ to denote the law of the particle driven by the SDE $\mathrm{d}\mathbf{X}_t = \boldsymbol{u}_t^\theta(\mathbf{X}_t)\mathrm{d}t + \sqrt{2\nu}\mathrm{d}\mathbf{B}_t$. To train $\boldsymbol{u}_t^\theta$, RDVN minimizes the loss $L(\theta) = \int_0^T \int_{\mathcal{X}} \|\boldsymbol{u}_t^\theta(\boldsymbol{x}) - K * \rho_t^\theta(\boldsymbol{x})\|_{\mathcal{L}^2}^2 \mathrm{d}\boldsymbol{x}\mathrm{d}t$. Note that in order to simulate the SDE, one needs to discretize the time variable in loss function $L$. After training $\theta$, RDVN outputs $\rho_t^\theta$ as a solution. However, no error estimation guarantee is provided that controls the discrepancy between $\rho_t^\theta$ and $\rho_t$ using $L(\theta)$.

Shen et al. [2022] propose the concept of self-consistency for the FPE. However, unlike our work where the EINN loss is derived via the stability analysis, they construct the potential $R(f)$ for the hypothesis velocity field $f$ by observing that the underlying velocity field $f^*$ is the fixed point of some velocity-consistent transformation $\mathcal{A}$ and they construct $R(f)$ to be a more complicated Sobolev norm of the residual $f - \mathcal{A}(f)$. In their result, they bound the Wasserstein distance between $\rho^f$ and $\rho$ by $R(f)$, which is weaker than our KL type control. The improved KL type control for the Fokker-Planck equation has also been discussed in [Boffi and Vanden-Eijnden, 2023]. A very recent work [Li et al., 2023] extends the self-consistency approach to compute the general Wasserstein gradient flow numerically, without providing further theoretical justification.

## 5 Experiments

To show the efficacy and efficiency of the proposed approach, we conduct numerical studies on example problems that admit explicit solutions and compare the results with SOTA NN-based PDE solvers. The included baselines are PINN [Raissi et al., 2019] and DRVN [Zhang et al., 2022]. Note that these baselines only considered the 2D NSE. We extend them to solve the MVE with the Coulomb interaction for comparison, and the details are discussed in Appendix C.1.

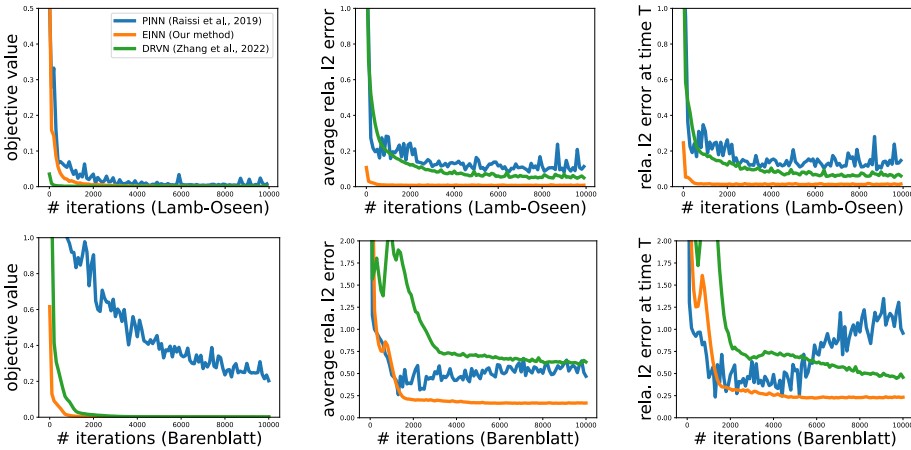

Figure 1: The first row contains results for the 2D NSE and the second row contains the results for the 3D MVE with Coulomb interaction. The first column reports the objective losses, while the second and third columns report the average and last-time-stamp relative $\ell_2$ error.

**Equations with an explicit solution**    We consider the following two instances that admit explicit solutions. We verify these solutions in Appendix C.2.

*Lamb-Oseen Vortex (2D NSE)* [Oseen, 1912]: Consider the whole domain case where $\mathcal{X} = \mathbb{R}^2$ and the Biot-Savart kernel (4). Let $\mathcal{N}(\boldsymbol{\mu}, \boldsymbol{\Sigma})$ be the Gaussian distribution with mean $\boldsymbol{\mu}$ and covariance $\boldsymbol{\Sigma}$. If $\rho_0 = \mathcal{N}(0, \sqrt{2\nu t_0}\boldsymbol{I}_2)$ for some $t_0 \geq 0$, then we have $\rho_t(\boldsymbol{x}) = \mathcal{N}(0, \sqrt{2\nu(t + t_0)}\boldsymbol{I}_2)$.

*Barenblatt solutions (MVE)* [Serfaty and Vázquez, 2014]: Consider the 3D MVE with the Coulomb interaction kernel (3) with the diffusion coefficient set to zero, i.e. $d = 3$ and $\nu = 0$. Let Uniform$[\mathbb{A}]$ be the uniform distribution over a set $\mathbb{A}$. Consider the whole domain case where $\mathcal{X} = \mathbb{R}^3$. If $\rho_0 = $ Uniform$[\|\boldsymbol{x}\| \leq (\frac{3}{4\pi}t_0)^{1/3}]$ for some $t_0 \geq 0$, then we have $\rho_t = $ Uniform$[\|\boldsymbol{x}\| \leq (\frac{3}{4\pi}(t + t_0))^{1/3}]$.

**Numerical results**    We present the results of our experiments in Figure 1, where the first row contains the result for the Lamb-Oseen vortex (2D NSE) and the second row contains the result for the Barenblatt model (3D MVE). The explicit solutions of these models allow us to assess the quality of the outputs of the included methods. Specifically, given a hypothesis solution $\rho_t^f$, the ground truth $\bar{\rho}_t$ and the interaction kernel $K$, define the relative $\ell_2$ error at timestamp $t$ as $Q(t) \overset{\text{def}}{=} \int_{\Omega} \|K * (\rho_t^f - \bar{\rho}_t)(\boldsymbol{x})\| / \|K * \bar{\rho}_t(\boldsymbol{x})\| d\boldsymbol{x}$, where $\Omega$ is some domain where $\rho_t$ has non-zero density. We are particularly interested in the quality of the convolution term $K * \rho_t^f$ since it has physical meanings. In the Biot-Savart kernel case, it is the velocity of the fluid, while in the Coulomb case, it is the Coulomb field. We set $\Omega$ to be $[-2, 2]^2$ for the Lamb-Oseen vortex and to $[-0.1, 0.1]^3$ for the Barenblatt model. For both models, we take $\nu = 0.1$, $t_0 = 0.1$, and $T = 1$. The neural network that we use is an MLP with 7 hidden layers, each of which has 20 neurons.

From the first column of Figure 1, we see that the objective loss of all methods has substantially reduced over a training period of 10000 iterations. This excludes the possibility that a baseline has worse performance because the NN is not well-trained, and hence the quality of the solution now solely depends on the efficacy of the method. From the second and third columns, we see that the proposed EINN method significantly outperforms the other two methods in terms of the time-average relative $\ell_2$ error, i.e. $\frac{1}{T}\int_0^T Q(t)dt$ and the relative $\ell_2$ error at the last time stamp $Q(T)$. This shows the advantage of of our method.

**Conclusion**    By employing entropy dissipation of the MVE, we design a potential function for a hypothesis velocity field such that it controls the KL divergence between the corresponding hypothesis solution and the ground truth, for any time stamp within the period of interest. Built on this potential, we proposed the EINN method for MVEs with NN and derived the detailed computation method of the stochastic gradient, using the classic adjoint method. Through empirical studies on examples of the 2D NSE and the 3D MVE with Coulomb interactions, we show the significant advantage of the proposed method, when compared with two SOTA NN based MVE solvers.

## Acknowledgments and Disclosure of Funding

Zhenfu Wang is supported by the National Key R&D Program of China, Project Number 2021YFA1002800, NSFC grant No.12171009, Young Elite Scientist Sponsorship Program by China Association for Science and Technology (CAST) No. YESS20200028 and the Fundamental Research Funds for the Central Universities (the start-up fund), Peking University. Zebang Shen's work is supported by ETH research grant and Swiss National Science Foundation (SNSF) Project Funding No. 200021-207343.

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
