in the field [Evans, 2022]. For the interest of this paper, we will only consider the methods that can be used to solve the MVE under consideration.

**Categorize PDE solvers via solution representation.**   To better understand the benefits of neural network (NN) based PDE solvers and to compare our approach with others, we categorize the literature based on the representation of the solution to the PDE. These representations can be roughly grouped into four categories:

- **1. Discretization-based representation:** The solution to the PDE is represented as discrete function values at grid points, finite-size cells, or finite-element meshes.
- **2. Representation as a combination of basis functions:** The solution to the PDE is approximated as a sum of basis functions, e.g. Fourier series, Legendre polynomials, or Chebyshev polynomials.
- **3. Representation using a collection of particles:** The solution to the PDE is represented as a collection of particles, each described by its weight, position, and other relevant information.
- **4. NN-based representation:** NNs offer many strategies for representing the solution to the PDE, such as using the NN directly to represent the solution, using normalizing flow or GAN-based parameterization to ensure the non-negativity and conservation of mass of the solution or using the NN to parameterize the underlying dynamics of the PDE, such as the time-varying velocity field that drives the evolution of the system.

The drawback of the first three strategies is that a sparse representation[2] leads to reduced solution accuracy, while a dense representation results in increased computational and memory cost. NNs, as powerful function approximation tools, are expected to surpass these strategies by being able to handle higher-dimensional, less regular, and more complicated systems [Weinan et al., 2021].

Given a representation strategy of the solution, an effective solver must exploit the underlying properties of the system to find the best candidate solution. Three-= notable properties that are utilized to design solvers are

- (A) the PDE definition or weak formulation of the system,
- (B) the SDE interpretation of the system,
- (C) the variational interpretation, particularly the Wasserstein gradient flow interpretation.

These properties are combined with the solution representations mentioned earlier to form different methods. For example, the Finite Difference method [Smith et al., 1985], Finite Volume method [Moukalled et al., 2016], and Finite Element method [Johnson, 2012] represent the solution of partial differential equations (PDEs) by discretizing the solution and utilize the property (A), at least in their original form. On the other hand, recent work by Carrillo et al. [2022] solves PDEs admitting a Wasserstein gradient flow structure using the classic JKO scheme [Jordan et al., 1998], which is based on the property (C), and the solution is also represented via discretization. The Spectral method [Shen et al., 2011] is a class of methods that exploits property (A) by representing the solution as a combination of basis functions. The Random Vortex Method [Long, 1988] is a highly successful method for solving the vorticity formulation of the 2D Navier-Stokes equation by exploiting property (C) and representing the solution with particles. The Blob method from Carrillo et al. [2019] is another particle-based method for solving PDEs that describe diffusion processes, which also exploits property (C).

# B  Comparison with Neural Operator

We thank the anonymous reviewers for pointing out the interesting research direction of neural operators [Xiao et al., 2023, Gupta et al., 2021, Li et al., 2020b, Kovachki et al., 2021, Li et al., 2020a]. However, to highlight the major difference between EINN and the approach of the Neural Operator,

---

[2]For example, sparser grid, cell or mesh with less granularity, fewer basis functions, fewer particles.

it's worth noting that they consider completely different problem settings: EINN requires *no pre-existing data* and the goal is to obtain the solution to a PDE by solely exploiting the structure of the equation itself. In contrast, the neural operator approach is *data-driven*, i.e. it relies on the existence of configuration-solution pairs. Here, by configuration-solution pairs, we mean the correspondence between some configurations that determine the PDE, e.g. the initial condition or the viscosity parameter in the fluid dynamics problems, and the pre-existing solution to the PDE given the aforementioned configurations. Consequently, the neural operator approach is more like a regression problem where a neural network is trained to learn the abstract map between the configuration and the solution. In contrast, EINN is more like a numerical PDE solver.

Consequently, EINN and the approach of neural operator are two related but quite distinct research directions. They are related in the sense that EINN can provide the data (configuration-solution pairs) required by the neural operator approach. They are distinct since EINN requires no data a priori, while the neural operator approach is built on the supervised learning paradigm.

## C   More Details on the Experiments

### C.1   Implementations of Baselines

**Objectives of PINN**

- For the vorticity equation of the 2D Navier-Stokes equation, let $\boldsymbol{u} : [0, T] \times \mathbb{R}^2 \to \mathbb{R}^2$ be the velocity field (this should not be confused with the velocity field of the continuity equation) such that $\nabla \cdot \boldsymbol{u} = 0$, i.e. $\boldsymbol{u}$ is divergence-free, and let $\omega = \nabla \times \boldsymbol{u} \in \mathbb{R}$ be the vorticity. We have

$$\frac{\partial \omega}{\partial t} + \nabla \cdot (\omega \boldsymbol{u}) = \nu \Delta \omega, \tag{24}$$

$$\omega = \nabla \times \boldsymbol{u}. \tag{25}$$

We use this form to construct the objective for the PINN method

$$\int_0^T \|\frac{\partial \omega}{\partial t} + \nabla \cdot (\omega \boldsymbol{u}) - \nu \Delta \omega\|_{\mathcal{L}(\Omega)^2}^2 + \|\omega - \nabla \times \boldsymbol{u}\|_{\mathcal{L}(\Omega)^2} dt, \tag{26}$$

where $\mathcal{L}^2(\Omega)$ denotes the functional $\mathcal{L}^2$ norm on the domain $\Omega = [-2, 2]^2$.

- For the MVE with Coulomb interaction, let $g$ be the Coulomb potential defined in equation 3. We have that $\psi = g * \rho$ is the solution to the Poisson equation $\Delta \psi = -\rho$ and $K * \rho = -\nabla \psi$. We have

$$\frac{\partial \rho}{\partial t} + \nabla \cdot (\rho \cdot (-\nabla \psi)) = \nu \Delta \rho \tag{27}$$

$$\Delta \psi = -\rho. \tag{28}$$

Expand the the divergence to obtain

$$\frac{\partial \rho}{\partial t} + \nabla \rho \cdot (-\nabla \psi) + \rho \cdot (-\Delta \psi) = \nu \Delta \rho \tag{29}$$

$$\Delta \psi = -\rho. \tag{30}$$

Now plug in the $\Delta \psi = -\rho$ to arrive at the following equivalent form

$$\frac{\partial \rho}{\partial t} + \nabla \rho \cdot -\nabla \psi + \rho^2 = \nu \Delta \rho \tag{31}$$

$$\Delta \psi = -\rho. \tag{32}$$

We use this form to construct the objective for the PINN method.

$$\int_0^T \|\frac{\partial \rho}{\partial t} + \nabla \rho \cdot -\nabla \psi + \rho^2 - \nu \Delta \rho\|_{\mathcal{L}^2(\Omega)}^2 + \|\Delta \psi + \rho\|_{\mathcal{L}^2}^2, \tag{33}$$

where $\mathcal{L}^2(\Omega)$ denotes the functional $\mathcal{L}^2$ norm on the domain $\Omega = [-1, 1]^2$.

**DRVN**   In the original paper [Zhang et al., 2022], only the Biot-Savart kernel is concerned. We extend the DRVN method to the Coulomb case by setting $K$ to be the kernel defined in equation 3.

## C.2 Examples with an Explicit Solution

In this section, we verify the explicit solutions discussed in the experiment section.

**Lamb-Oseen Vortex on the whole domain $\mathbb{R}^2$.** Recall that we consider the 2D Navier-Stokes equation (the MVE with the Biot-Savart interaction kernel (4)). The Lamb-Oseen Vortex model states that, if $\rho_0 = \mathcal{N}(0, \sqrt{2\nu t_0} \boldsymbol{I}_2)$ for some $t_0 \geq 0$, then we have $\rho_t(\boldsymbol{x}) = \mathcal{N}(0, \sqrt{2\nu(t+t_0)} \boldsymbol{I}_2)$.

To verify this, define $\boldsymbol{u}_t(\boldsymbol{x}) = \frac{1}{\sqrt{\nu(t+t_0)}} \boldsymbol{v}\left(\frac{\boldsymbol{x}}{\sqrt{\nu(t+t_0)}}\right)$, where

$$\boldsymbol{v}(\boldsymbol{x}) = \frac{1}{2\pi} \frac{\boldsymbol{x}^\perp}{\|\boldsymbol{x}\|^2} \left(1 - \exp(-\frac{1}{4}\|\boldsymbol{x}\|^2)\right). \tag{34}$$

One can easily check that $\nabla \cdot \boldsymbol{u}_t \equiv 0$ and hence there exists a function $\psi_t$ such that $\nabla^\perp \psi_t = -\boldsymbol{u}_t$, where $\nabla^\perp$ denotes the perpendicular gradient, defined as $\nabla^\perp = (-\partial_{\boldsymbol{x}_2}, \partial_{\boldsymbol{x}_2})$, and $\psi_t$ is called the stream function in the literature of fluid dynamics. Moreover, one can verify that $\nabla \times \boldsymbol{u}_t = \rho_t$ where $\nabla \times$ denotes the curl of a 2D velocity field, defined as $\nabla \times \boldsymbol{u}(\boldsymbol{x}) = \partial u_2 / \partial x_1 - \partial u_1 / \partial x_2$. Together we have

$$\Delta \psi_t = -\rho_t, \tag{35}$$

i.e., the stream function $\psi_t$ is the solution to the 2D Poisson equation with a source term $\rho_t$.

Under the boundary condition that $\psi_t(\boldsymbol{x}) \to 0$ for $\|\boldsymbol{x}\| \to \infty$, we can express $\psi_t$ via the unique Green function $G(\boldsymbol{x}) = \frac{1}{2\pi} \ln \|\boldsymbol{x}\|$ as

$$\psi_t(\boldsymbol{x}) = G * \rho_t = \frac{1}{2\pi} \int \ln \|\boldsymbol{x} - \boldsymbol{y}\| \rho_t(\boldsymbol{y}) d\boldsymbol{y}. \tag{36}$$

Consequently, by taking the perpendicular gradient, we obtain

$$\boldsymbol{u}_t = \nabla^\perp \psi_t = \frac{1}{2\pi} \int \frac{(\boldsymbol{x} - \boldsymbol{y})^\perp}{\|\boldsymbol{x} - \boldsymbol{y}\|^2} \rho_t(\boldsymbol{y}) d\boldsymbol{y} = K * \rho_t. \tag{37}$$

Finally, by plugging the expressions of $\rho_t$ and $\boldsymbol{u}_t = K * \rho_t$ in the MVE (2), we verified the Lamb-Oseen vortex.

**Barenblatt solutions for the MVE with Coulomb Interaction.** Recall that we consider the MVE with the Coulomb interaction kernel (3) for $d = 3$ and set the diffusion coefficient $\nu = 0$, i.e.

$$\frac{\partial \rho_t}{\partial t} + \nabla \cdot (\rho_t \cdot -\nabla \psi_t) = 0 \tag{38}$$

where $\psi_t$ is the solution to the Poisson equation $\Delta \psi_t = -\rho_t$. The Barenblatt solution of the above MVE is stated as follows: If $\rho_0 = \text{Uniform}\left[\|\boldsymbol{x}\| \leq \left(\frac{3}{4\pi} t_0\right)^{1/3}\right]$ for some $t_0 \geq 0$, then we have

$$\rho_t = \text{Uniform}\left[\|\boldsymbol{x}\| \leq \left(\frac{3}{4\pi}(t + t_0)\right)^{1/3}\right] \tag{39}$$

We now verify this solution.

Recall that the volume of a three dimensional Euclidean ball with radius $R$ is $\frac{4\pi}{3} R^3$. Hence we can write the density function as $\rho_t(x) = \frac{1}{t+t_0} \chi_{\|\boldsymbol{x}\| \leq (\frac{3}{4\pi}(t+t_0))^{1/3}}$, where $\chi_{\mathbb{X}}$ is a function that takes value 1 for $\boldsymbol{x} \in \mathbb{X}$ and takes value 0 for $\boldsymbol{x} \notin \mathbb{X}$. Take

$$\psi_t(\boldsymbol{x}) = \begin{cases} \frac{2(\frac{3}{4\pi}(t+t_0))^{2/3} - \|\boldsymbol{x}\|^2}{6(t+t_0)}, & \|\boldsymbol{x}\| \leq \left(\frac{3}{4\pi}(t+t_0)\right)^{1/3}, \\ \frac{1}{8\pi\|\boldsymbol{x}\|}, & \|\boldsymbol{x}\| > \left(\frac{3}{4\pi}(t+t_0)\right)^{1/3}. \end{cases} \tag{40}$$

It can be verified that the Poisson equation $\Delta \psi_t = -\rho_t$ holds (note that $\Delta \|\boldsymbol{x}\|^{-1} = 0$, i.e. $\|\boldsymbol{x}\|^{-1}$ is a harmonic function for $d = 3$). Consequently, for a fixed time stamp $t$ and any $\|\boldsymbol{x}\| \leq \left(\frac{3}{4\pi}(t+t_0)\right)^{1/3}$ we have

$$\frac{\partial \rho_t}{\partial t}(\boldsymbol{x}) + \nabla \cdot (\rho_t(\boldsymbol{x}) \cdot -\nabla \psi_t(\boldsymbol{x})) = -\frac{1}{(t+t_0)^2} + \frac{1}{(t+t_0)^2} = 0, \tag{41}$$

which verifies this solution.

## D  Adjoint Method

Consider the ODE system

$$\dot{s}(t) = \psi(s(t), t, \theta)$$
$$s(0) = s_0,$$

and the objective loss

$$\ell(\theta) = \int_0^T g(s(t), t, \theta)\mathrm{d}t. \tag{42}$$

The following proposition computes the gradient of $\ell$ w.r.t. $\theta$. We omit the parameters of the functions for succinctness. We note that all the functions in the integrands should be evaluated at the corresponding time stamp $t$, e.g. $b^\top \frac{\partial h}{\partial \theta}\mathrm{d}t$ abbreviates for $b(t)^\top \frac{\partial}{\partial \theta} h(\xi(t), x(t), t, \theta)\mathrm{d}t$.

**Proposition 1.**

$$\frac{\mathrm{d}\ell}{\mathrm{d}\theta} = \int_0^T a^\top \frac{\partial \psi}{\partial \theta} + \frac{\partial g}{\partial \theta}\mathrm{d}t. \tag{43}$$

*where $a(t)$ is solution to the following final value problems*

$$\dot{a}^\top + a^\top \frac{\partial \psi}{\partial s} + \frac{\partial g}{\partial s} = 0, a(T) = 0, \tag{44}$$

*Proof.* Let us define the Lagrange multiplier function (or the adjoint state) $a(t)$ dual to $s(t)$. Moreover, let $L$ be an augmented loss function of the form

$$L = \ell - \int_0^T a^\top (\dot{s} - \psi)\mathrm{d}t. \tag{45}$$

Since we have $\dot{s}(t) = \psi(s(t), t, \theta)$ by construction, the integral term in $L$ is always null and $a$ can be freely assigned while maintaining $\mathrm{d}L/\mathrm{d}\theta = \mathrm{d}\ell/\mathrm{d}\theta$. Using integral by part, we have

$$\int_0^T a^\top \dot{s} \,\mathrm{d}t = a(t)^\top s(t)|_0^T - \int_0^T s^\top \dot{a} \,\mathrm{d}t. \tag{46}$$

We obtain

$$L = -a(t)^\top s(t)|_0^T + \int_0^T \dot{a}^\top s + a^\top \psi + g \,\mathrm{d}t. \tag{47}$$

Now we compute the gradient of $L$ w.r.t. $\theta$ as

$$\frac{\mathrm{d}\ell}{\mathrm{d}\theta} = \frac{\mathrm{d}L}{\mathrm{d}\theta} = -a(T)^\top \frac{\mathrm{d}x(T)}{\mathrm{d}\theta} + \int_0^T \dot{a}^\top \frac{\mathrm{d}s}{\mathrm{d}\theta} + a^\top \left( \frac{\partial \psi}{\partial \theta} + \frac{\partial \psi}{\partial s}\frac{\mathrm{d}s}{\mathrm{d}\theta} \right)\mathrm{d}t + \int_0^T \frac{\partial g}{\partial s}\frac{\mathrm{d}s}{\mathrm{d}\theta} + \frac{\partial g}{\partial \theta}\mathrm{d}t,$$

which by rearranging terms yields to

$$\frac{\mathrm{d}\ell}{\mathrm{d}\theta} = \frac{\mathrm{d}L}{\mathrm{d}\theta} = -a(T)^\top \frac{\mathrm{d}x(T)}{\mathrm{d}\theta} + \int_0^T a^\top \frac{\partial \psi}{\partial \theta} + \frac{\partial g}{\partial \theta}\mathrm{d}t + \int_0^T \left( \dot{a}^\top + a^\top \frac{\partial \psi}{\partial s} + \frac{\partial g}{\partial s} \right)\frac{\mathrm{d}s}{\mathrm{d}\theta}\mathrm{d}t.$$

Now by taking $a$ satisfying the *final* value problems

$$\dot{a}^\top + a^\top \frac{\partial \psi}{\partial s} + \frac{\partial g}{\partial s} = 0, a(T) = 0, \tag{48}$$

we derive the result

$$\frac{\mathrm{d}\ell}{\mathrm{d}\theta} = \int_0^T a^\top \frac{\partial \psi}{\partial \theta} + \frac{\partial g}{\partial \theta}\mathrm{d}t. \tag{49}$$

### D.1  Writing the Trajectory-wise Loss (15) in an ODE-constrained form

We are now ready to write $R(f_\theta; x_0)$ in an ODE-constrained form. Define the state $s(t)$, the initial condition $s_0$ and the transition function $\psi$ as follows: Let

$$s(t) = \left[ x(t), \xi(t), \{y_i(t)\}_{i=1}^N, \{\zeta_i(t)\}_{i=1}^N \right], \tag{50}$$

with $\xi(t) = \nabla \log \rho_t^f(\boldsymbol{x}(t))$ and $\zeta_i(t) = \nabla \log \rho_t^f(\boldsymbol{y}_i(t))$. Take the initial condition

$$\boldsymbol{s}_0 = \left[\boldsymbol{x}_0, \xi_0, \{\boldsymbol{y}_i(0)\}_{i=1}^N, \{\zeta_i(0)\}_{i=1}^N\right] \tag{51}$$

with $\xi_0 = \nabla \log \bar{\rho}_0(\boldsymbol{x}_0)$, $\zeta_i(0) = \nabla \log \bar{\rho}_0(\boldsymbol{y}_i(0))$, and $\boldsymbol{y}_i(0) \overset{iid}{\sim} \bar{\rho}_0$; and define the function

$$\psi(t, s(t); \theta) = [f_t(\boldsymbol{x}(t); \theta), \; h_t(\boldsymbol{x}(t), \xi(t); \theta), \; \{f_t(\boldsymbol{y}_i(t); \theta)\}_{i=1}^N, \; \{h_t(\boldsymbol{y}_i(t), \zeta_i(t); \theta)\}_{i=1}^N], \tag{52}$$

where $h(\boldsymbol{a}, \boldsymbol{b}; \theta) = -\nabla\left(\operatorname{div} f_t(\boldsymbol{a}; \theta)\right) - \mathcal{J}_{f_t}^\top(\boldsymbol{a}; \theta)\boldsymbol{b}$ (derived from equation 18). Finally, define

$$g(t, \boldsymbol{s}(t); \theta) = \|f(t, \boldsymbol{x}(t); \theta) - (-\nabla V(\boldsymbol{x}(t)) + E(t, \boldsymbol{s}(t)) - \nu\xi(t))\|^2, \tag{53}$$

where the estimator $E(t, \boldsymbol{s})$ of the convolution term is defined as

$$E(t, \boldsymbol{s}(t)) = \begin{cases} \frac{1}{N} \sum_{i=1}^N K_c(\boldsymbol{x}(t) - \boldsymbol{y}_i(t)) & \text{the Coulomb case,} \\ \frac{1}{N} \sum_{i=1}^N U(\boldsymbol{x} - \boldsymbol{y}_i(t))\zeta_i(t) & \text{the Biot-Savart case.} \end{cases} \tag{54}$$

We recall the definition of $U$ in equation 21 and the definition of $K_c$ in equation 19. $\qquad\square$

# E    Detailed Proofs

*Proof of Lemma 1.* Recall the McKean-Vlasov equation 6 and the continuity equation 10. We simply write that $\rho_t = \rho_t^f$ and omit the integration domain $\mathcal{X}$. Then

$$
\frac{\mathrm{d}}{\mathrm{d}t} \int \rho_t \log \frac{\rho_t}{\bar{\rho}_t} = \int \partial_t \rho_t \log \frac{\rho_t}{\bar{\rho}_t} + \int \rho_t \partial_t \log \rho_t - \int \rho_t \partial_t \log \bar{\rho}_t
$$

$$
= - \int \mathrm{div}\Big(\rho_t\Big(\big[-\nabla V(x) + K * \rho_t - \nu \nabla \log \rho_t\big] + \delta_t\Big)\Big) \log \frac{\rho_t}{\bar{\rho}_t}
$$

$$
+ \int \frac{\rho_t}{\bar{\rho}_t} \mathrm{div}\Big(\bar{\rho}_t\big(-\nabla V(x) + K * \bar{\rho}_t - \nu \nabla \log \bar{\rho}_t\big)\Big),
$$

where we note that $\int \rho_t \partial_t \log \rho_t = \int \partial_t \rho_t = 0$ since the total mass is preserved over time. By integration by parts, one has

$$
\frac{\mathrm{d}}{\mathrm{d}t} \int \rho_t \log \frac{\rho_t}{\bar{\rho}_t} = I_1 + I_2 + I_3 + \int \rho_t \delta_t \cdot \nabla \log \frac{\rho_t}{\bar{\rho}_t},
$$

where $I_1, I_2, I_3$ denote the linear, nonlinear interaction, and diffusion parts separately. More precisely, by integration by parts,

$$
I_1 = \int \mathrm{div}(\rho_t \nabla V(x)) \log \frac{\rho_t}{\bar{\rho}_t} - \int \frac{\rho_t}{\bar{\rho}_t} \mathrm{div}(\bar{\rho}_t \nabla V(x))
$$

$$
= - \int \rho_t \nabla V(x) \cdot \nabla \log \frac{\rho_t}{\bar{\rho}_t} + \int \bar{\rho}_t \nabla \frac{\rho_t}{\bar{\rho}_t} \cdot \nabla V(x) = 0.
$$

And

$$
I_2 = - \int \mathrm{div}(\rho_t K * \rho_t) \log \frac{\rho_t}{\bar{\rho}_t} + \int \frac{\rho_t}{\bar{\rho}_t} \mathrm{div}(\bar{\rho}_t K * \bar{\rho}_t)
$$

$$
= \int \rho_t K * \rho_t \nabla \log \frac{\rho_t}{\bar{\rho}_t} - \int \bar{\rho}_t K * \bar{\rho}_t \cdot \nabla \frac{\rho_t}{\bar{\rho}_t}
$$

$$
= \int \rho_t \nabla \log \frac{\rho_t}{\bar{\rho}_t} \cdot K * (\rho_t - \bar{\rho}_t).
$$

Given that the kernel $K$ is divergence free, that is $\mathrm{div} K = 0$, one further has

$$
I_2 = - \int \rho_t \nabla \log \bar{\rho}_t \cdot K * (\rho_t - \bar{\rho}_t) + \int \nabla \rho_t \cdot K * (\rho_t - \bar{\rho}_t)
$$

$$
= - \int \rho_t \nabla \log \bar{\rho}_t \cdot K * (\rho_t - \bar{\rho}_t).
$$

$$(55)$$

Note that this modification will be used in the proof in the 2D Navier-Stokes case. Finally, all diffusion terms sum up to $I_3$ which can be further simplified as

$$
I_3 = \nu \int \mathrm{div}(\rho_t \nabla \log \rho_t) \log \frac{\rho_t}{\bar{\rho}_t} - \nu \int \frac{\rho_t}{\bar{\rho}_t} \mathrm{div}(\bar{\rho}_t \nabla \log \bar{\rho}_t)
$$

$$
= -\nu \int \rho_t \nabla \log \rho_t \cdot \nabla \log \frac{\rho_t}{\bar{\rho}_t} + \nu \int \bar{\rho}_t \nabla \log \bar{\rho}_t \cdot \nabla \frac{\rho_t}{\bar{\rho}_t}
$$

$$
= -\nu \int \rho_t |\nabla \log \frac{\rho_t}{\bar{\rho}_t}|^2.
$$

We thus complete the proof of Lemma 1.

$\square$

*Proof of Lemma 2.* Recall that $K = -\nabla g$. For simplicity, we write that $\rho_t = \rho_t^f$. Then

$$
\frac{\mathrm{d}}{\mathrm{d}t} F(\rho_t, \bar{\rho}_t) = \frac{\mathrm{d}}{\mathrm{d}t} \frac{1}{2} \int_{\mathcal{X}^2} g(x-y) \mathrm{d}(\rho_t - \bar{\rho}_t)^{\otimes 2}(x, y)
$$

$$
= \int_{\mathcal{X}} g * (\rho_t - \bar{\rho}_t)(x)\big(\partial_t \rho_t(x) - \partial_t \bar{\rho}_t(x)\big) \mathrm{d}x
$$

$$
= \int g * (\rho_t - \bar{\rho}_t)(x) \mathrm{div}\Big\{\rho_t\big([\nabla V(x) - K * \rho_t + \nu \nabla \log \rho_t] - \delta_t\big)
$$

$$
- \bar{\rho}_t\big(\nabla V(x) - K * \bar{\rho}_t + \nu \log \bar{\rho}_t\big)\Big\}
$$

$$
= J_1 + J_2 + J_3 + J_4,
$$

where $J_1, J_2, J_3, J_4$ denote the perturbation term, the linear difference term, the nonlinear difference term, and the diffusion term respectively. The perturbation term $J_1$ reads

$$J_1 = -\int_{\mathcal{X}} g * (\rho_t - \bar{\rho}_t) \operatorname{div}(\rho_t \delta_t) = -\int_{\mathcal{X}} \rho_t K * (\rho_t - \bar{\rho}_t) \cdot \delta_t.$$

By integration by parts, the linear difference term can be written as

$$J_2 = \int_{\mathcal{X}} g * (\rho_t - \bar{\rho}_t) \operatorname{div}\Big((\rho_t - \bar{\rho}_t)\nabla V\Big) = \int_{\mathcal{X}} K * (\rho_t - \bar{\rho}_t)(\rho_t - \bar{\rho}_t)\nabla V$$

$$= \frac{1}{2}\int_{\mathcal{X}^2} K(x-y)(\nabla V(x) - \nabla V(y)) \mathrm{d}(\rho_t - \bar{\rho}_t)^{\otimes 2}(x,y),$$

where the last equality is true since $K = -\nabla g$ is an odd function and we do the symmetrization trick, i.e. exchanging the role of $x$ and $y$ to another term and then taking the average.

The nonlinear difference term reads

$$J_3 = -\int_{\mathcal{X}} g * (\rho_t - \bar{\rho}_t) \operatorname{div}\Big(\rho_t K * \rho_t - \bar{\rho}_t K * \bar{\rho}_t\Big)$$

$$= -\int_{\mathcal{X}} K * (\rho_t - \bar{\rho}_t)(\rho_t K * (\rho_t - \bar{\rho}_t)) - \int_{\mathcal{X}} K * (\rho_t - \bar{\rho}_t)(\rho_t - \bar{\rho}_t)K * \bar{\rho}_t$$

$$= -\int_{\mathcal{X}} \rho_t |K * (\rho_t - \bar{\rho}_t)|^2 - \frac{1}{2}\int K(x-y)(K * \bar{\rho}_t(x) - K * \bar{\rho}_t(y)) \mathrm{d}(\rho_t - \bar{\rho}_t)^{\otimes 2}(x,y),$$

where again in the last term we do the symmetrization.

The diffusion term reads

$$J_4 = \nu \int g * (\rho_t - \bar{\rho}_t) \operatorname{div}\Big(\rho_t \nabla \log \rho_t - \bar{\rho}_t \nabla \log \bar{\rho}_t\Big)$$

$$= \nu \int K * (\rho_t - \bar{\rho}_t)\rho_t \nabla \log \frac{\rho_t}{\bar{\rho}_t} + \nu \int K * (\rho_t - \bar{\rho}_t)(\rho_t - \bar{\rho}_t)\nabla \log \bar{\rho}_t$$

$$= \nu \int_{\mathcal{X}} \rho_t K * (\rho_t - \bar{\rho}_t) \cdot \nabla \log \frac{\rho_t}{\bar{\rho}_t}$$

$$\quad + \frac{\nu}{2}\int_{\mathcal{X}^2} K(x-y)(\nabla \log \bar{\rho}_t(x) - \nabla \log \bar{\rho}_t(y)) \mathrm{d}(\rho_t - \bar{\rho}_t)^{\otimes 2}.$$

To sum it up, we prove the thesis.

$\square$

## E.1 Proof of the 2D Navier-Stokes case

Now we proceed to control the growth of the KL divergence $\mathbf{KL}(\rho_t^f|\bar{\rho}_t)$ for the 2D Navier-Stokes case. Since the Biot-Savart law is divergence free, by equation 55 in the proof of Lemma 1, one has

$$\frac{\mathrm{d}}{\mathrm{d}t}\int_{\Pi^d} \rho_t \log \frac{\rho_t}{\bar{\rho}_t} = -\nu \int_{\Pi^d} \rho_t |\nabla \log \frac{\rho_t}{\bar{\rho}_t}|^2 - \int_{\Pi^d} \rho_t K * (\rho_t - \bar{\rho}_t) \cdot \nabla \log \bar{\rho}_t + \int_{\Pi^d} \rho_t \delta_t \cdot \nabla \log \frac{\rho_t}{\bar{\rho}_t}. \quad (56)$$

Recall that we write the kernel $K = (K_1, \cdots, K_d)$ and its component $K_i = \sum_{j=1}^d \partial_{x_j} U_{ij}(x)$, where $U = (U_{ij})_{1 \le i,j \le d}$ is a matrix-valued potential function for instance can be defined as in equation 21. Consequently

$$-\int \rho_t K * (\rho_t - \bar{\rho}_t) \cdot \nabla \log \bar{\rho}_t = -\sum_{i,j=1}^d \int \rho_t \partial_{x_j} U_{ij} * (\rho_t - \bar{\rho}_t)\partial_{x_i} \log \bar{\rho}_t,$$

which equals to

$$\sum_{i,j=1}^d \int U_{ij} * (\rho_t - \bar{\rho}_t)\partial_{x_j}\Big(\frac{\rho_t}{\bar{\rho}_t}\partial_{x_i}\bar{\rho}_t\Big) = A + B$$

by integration by parts, where further

$$A = \sum_{i,j=1}^d \int V_{ij} * (\rho_t - \bar{\rho}_t) \, \partial_{x_i}\bar{\rho}_t \, \partial_{x_j} \frac{\rho_t}{\bar{\rho}_t} = \int U * (\rho_t - \bar{\rho}_t) : \nabla \bar{\rho}_t \otimes \nabla \frac{\rho_t}{\bar{\rho}_t},$$

and
$$B = \sum_{i,j=1}^{d} \int \rho_t U_{ij} * (\rho_t - \bar{\rho}_t) \frac{\partial^2_{x_i x_j} \bar{\rho}_t}{\bar{\rho}_t} = \int \rho_t U * (\rho_t - \bar{\rho}_t) : \frac{\nabla^2 \bar{\rho}_t}{\bar{\rho}_t}.$$

Noticing that $\nabla \frac{\rho_t}{\bar{\rho}_t} = \frac{\rho_t}{\bar{\rho}_t} \nabla \log \frac{\rho_t}{\bar{\rho}_t}$, one estimates $A$ as follows

$$A = \int \rho_t U * (\rho_t - \bar{\rho}_t) : \nabla \log \bar{\rho}_t \otimes \nabla \log \frac{\rho_t}{\bar{\rho}_t}$$

$$\le \frac{\nu}{4} \int \rho_t |\nabla \log \frac{\rho_t}{\bar{\rho}_t}|^2 + \frac{1}{\nu} \int \rho_t |(\nabla \log \bar{\rho}_t)^\top U * (\rho - \bar{\rho})|^2$$

$$\le \frac{\nu}{4} \int \rho_t |\nabla \log \frac{\rho_t}{\bar{\rho}_t}|^2 + \frac{1}{\nu} \|U\|_{L^\infty}^2 \|\nabla \log \bar{\rho}_t\|_{L^\infty}^2 \|\rho_t - \bar{\rho}_t\|_{L^1}^2,$$

and again by Csiszár–Kullback–Pinsker inequality, one has that

$$A \le \frac{\nu}{4} \int \rho_t |\nabla \log \frac{\rho_t}{\bar{\rho}_t}|^2 + \frac{2}{\nu} \|U\|_{L^\infty}^2 \|\nabla \log \bar{\rho}_t\|_{L^\infty}^2 \int \rho_t \log \frac{\rho_t}{\bar{\rho}_t}.$$

Now it only remains to control $B$. Recall the following famous Gibbs inequality

**Lemma 4** (Gibbs inequality). *For any parameter $\eta > 0$, and probability measures $\rho, \bar{\rho} \in \mathcal{P}(\mathcal{X}) \cap L^1(\mathcal{X})$, and $\phi$ a real-valued function defined on $\mathcal{X}$, one has the following change of reference measure inequality*

$$\int_{\mathcal{X}} \rho(x)\phi(x)\mathrm{d}x \le \frac{1}{\eta} \Big( \int_{\mathcal{X}} \rho(x) \log \frac{\rho(x)}{\bar{\rho}(x)} \mathrm{d}x + \log \int_{\mathcal{X}} \bar{\rho}(x) \exp(\eta\phi(x))\mathrm{d}x \Big).$$

The proof of this inequality can be found in section 13.1 in [Erdős and Yau, 2017].

To control $B$, we write that $\phi = U * (\rho_t - \bar{\rho}_t) : \frac{\nabla^2 \bar{\rho}_t}{\bar{\rho}_t}$ and thus $B = \int \rho_t \phi$. We choose a positive parameter $\eta > 0$ such that

$$\frac{1}{\eta} = 2\|U\|_{L^\infty} \Big\| \frac{\nabla^2 \bar{\rho}_t}{\bar{\rho}_t} \Big\|_{L^\infty}.$$

Now we apply Lemma 4 to obtain that

$$B = \int \rho_t \phi \le \frac{1}{\eta} \left( \int \rho_t \log \frac{\rho_t}{\bar{\rho}_t} + \log \int \bar{\rho}_t \exp(\eta\phi) \right).$$

Note that $\eta > 0$ is chosen so small such that

$$\eta \|\phi\|_{L^\infty} \le \frac{1}{2\|U\|_{L^\infty} \Big\| \frac{\nabla^2 \bar{\rho}_t}{\bar{\rho}_t} \Big\|_{L^\infty}} \|U\|_{L^\infty} \|\rho_t - \bar{\rho}_t\|_{L^1} \Big\| \frac{\nabla^2 \bar{\rho}_t}{\bar{\rho}_t} \Big\|_{L^\infty}$$

$$\le \frac{1}{2} \|\rho_t - \bar{\rho}_t\|_{L^1} \le 1,$$

since for two probability densities it always holds $\|\rho_t - \bar{\rho}_t\|_{L^1} \le 2$. Consequently, applying the inequality $\exp(x) \le 1 + x + \frac{e}{2}x^2$ for $|x| \le 1$, we have

$$\int \bar{\rho}_t \exp(\eta\phi) \le \int \bar{\rho}_t \left( 1 + \eta\phi + \frac{e}{2}\eta^2\phi^2 \right) \le 1 + 0 + \frac{e}{2} \left( \frac{1}{2} \|\rho_t - \bar{\rho}_t\|_{L^1} \right)^2 \le 1 + \frac{e}{4} \mathbf{KL}(\rho_t|\bar{\rho}_t),$$

where

$$\int \bar{\rho}_t \phi = \int U * (\rho_t - \bar{\rho}_t) : \nabla^2 \bar{\rho}_t = \int \sum_{i,j=1}^{d} \partial_{x_i x_j} U * (\rho_t - \bar{\rho}_t) \bar{\rho}_t = \int \mathrm{div} K * (\rho_t - \bar{\rho}_t) \bar{\rho}_t = 0,$$

since $\mathrm{div} K = 0$.

To sum it up, in particular since $\log(1 + x) \le x$ for $x > 0$, one has

$$B \le \frac{1}{\eta} \Big( 1 + \frac{e}{4} \Big) \mathbf{KL}(\rho_t|\bar{\rho}_t) \le 4\|U\|_{L^\infty} \Big\| \frac{\nabla^2 \bar{\rho}_t}{\bar{\rho}_t} \Big\|_{L^\infty} \mathbf{KL}(\rho_t|\bar{\rho}_t).$$

Combining equation 56, the estimates for $A$ and $B$, one has

$$\frac{\mathrm{d}}{\mathrm{d}t} \int \rho_t \log \frac{\rho_t}{\bar{\rho}_t} \le -\frac{3\nu}{4} \int \rho_t |\nabla \log \frac{\rho_t}{\bar{\rho}_t}|^2 + M(t) \int \rho_t \log \frac{\rho_t}{\bar{\rho}_t} + \int \rho_t \delta_t \cdot \nabla \log \frac{\rho_t}{\bar{\rho}_t}$$

$$\le -\frac{\nu}{2} \int \rho_t |\nabla \log \frac{\rho_t}{\bar{\rho}_t}|^2 + M(t) \int \rho_t \log \frac{\rho_t}{\bar{\rho}_t} + \frac{1}{\nu} \int \rho_t |\delta_t|^2 \tag{57}$$

where

$$M(t) = \frac{2}{\nu} \|U\|_{L^\infty}^2 \|\nabla \log \bar{\rho}_t\|_{L^\infty}^2 + 4\|U\|_{L^\infty} \left\| \frac{\nabla^2 \bar{\rho}_t}{\bar{\rho}_t} \right\|_{L^\infty} = M(t; \nu, U, \bar{\rho}_t).$$

Since the matrix-valued potential function $U$ is bounded ($\|U(\boldsymbol{x})\|_{op} \le 1/4$ when $U$ takse the form (21)), and under suitable assumptions for the initial data $\bar{\rho}_0$ (for instance $\bar{\rho}_0 \in C^3$ and there exists $c > 1$ s.t. $\frac{1}{c} \le \bar{\rho} \le c$), one can obtain $\sup_{t\in[0,T]} M(t) \le M < \infty$. We recall Theorem 2 in [Guillin et al., 2021] as below for completeness.

**Theorem 5.** *Given the initial data $\bar{\rho}_0 \in C^\infty(\Pi^d)$, such that there exists $c > 1$, $\frac{1}{c} \le \bar{\rho}_0 \le c$. Then the vorticity formulation of the 2D Navier-Stokes equation*

$$\partial_t \bar{\rho}_t + \mathrm{div}(\bar{\rho}_t K * \bar{\rho}_t) = \nu \Delta \bar{\rho}_t, \quad \bar{\rho}(0, x) = \bar{\rho}_0(x),$$

*has a unique bounded solution $\bar{\rho}(t, x) \in C^\infty([0, \infty) \times \Pi^d)$, and for any $t > 0$, for any $x \in \Pi^d$, it holds that $\frac{1}{c} \le \bar{\rho}(t, x) \le c$.*

Finally, we simplify equation 57 to obtain that

$$\frac{\mathrm{d}}{\mathrm{d}t} \int \rho_t \log \frac{\rho_t}{\bar{\rho}_t} \le M \int \rho_t \log \frac{\rho_t}{\bar{\rho}_t} + \frac{1}{\nu} \int \rho_t |\delta_t|^2,$$

where $M = \sup_{t\in[0,T]} M(t; \nu, U, \bar{\rho}_t) < \infty$. By Gronwall inequality, one finally obtains that

$$\sup_{t\in[0,T]} \int_{\Pi^d} \rho_t \log \frac{\rho_t}{\bar{\rho}_t} \mathrm{d}x \le \frac{1}{\nu} \exp(MT) R(\theta).$$

As noted in [Guillin et al., 2021],in particular Corollary 2 there, one can improve the above time-dependent estimate $(\exp(MT))$ to uniform-in-time estimate by using Logarithmic Sobolev inequality. Indeed, given that $\frac{1}{c} \le \bar{\rho}_t \le c$, one has that

$$\int_{\Pi^d} \rho_t \log \frac{\rho_t}{\bar{\rho}_t} \mathrm{d}x \le \frac{c^2}{8\pi^2} \int_{\Pi^d} \rho_t |\nabla_x \log \frac{\rho_t}{\bar{\rho}_t}|^2 \mathrm{d}x. \tag{58}$$

Combining equation 58 and equation 57, one obtains that

$$\frac{\mathrm{d}}{\mathrm{d}t} \mathbf{KL}(\rho_t|\bar{\rho}_t) \le \left( M(t) - \frac{4\pi^2 \nu}{c^2} \right) \mathbf{KL}(\rho_t|\bar{\rho}_t) + \frac{1}{\nu} \int \rho_t |\delta_t|^2.$$

Multiplying the factor $\exp\left( \frac{4\pi^2 \nu}{c^2} t - \int_0^t M(s)\mathrm{d}s \right)$ and noting in particular $\mathbf{KL}(\rho_0|\bar{\rho}_0) = 0$, one obtains that

$$\mathbf{KL}(\rho_t|\bar{\rho}_t) \le \int_0^t \exp\left( \frac{4\pi^2 \nu}{c^2}(s - t) + \int_s^t M(u)\mathrm{d}u \right) f(s)\mathrm{d}s.$$

Indeed, under the assumptions as in Theorem 2, one has that there exists a universal $C > 0$, such that

$$\int_0^\infty M(t)\mathrm{d}t = C < \infty.$$

We thus immediately obtain that

$$\sup_{t\in[0,T]} \mathbf{KL}(\rho_t|\bar{\rho}_t) \le \frac{e^C}{\nu} \int_0^T \int_{\Pi^d} \rho_t |\delta_t|^2 \mathrm{d}x \mathrm{d}t.$$

This completes the proof of Theorem 2.

**The McKean-Vlasov PDEs, i.e. equation 2, with bounded interactions** $K \in L^\infty$    As mentioned in the main body of this article, it is much easier to obtain the stability estimate for the McKean-Vlasov PDE with bounded interactions.

**Theorem 6** (Stability Estimate for McKean-Vlasov PDE with $K \in L^\infty$). *Assume that $K \in L^\infty$. One has the estimate that*

$$\sup_{t \in [0,T]} \mathbf{KL}(\rho_t^f | \bar{\rho}_t) \leq \frac{1}{\nu} \exp\Big(\frac{2\|K\|_{L^\infty}^2}{\nu} T\Big) R(f),$$

*where we recall the self-consitency potential/loss function $R(\theta)$ reads*

$$R(f) = \int_0^T \int_{\mathcal{X}} |f(t,x) + \nabla V(x) - K * \rho_t^f + \nu \nabla \log \rho_t^\theta|^2 \mathrm{d}\rho_t^f(x)\mathrm{d}t.$$

*Proof.* Here we give the control of the growth of the KL divergence for systems with bounded kernels. Applying Cauchy-Schwarz inequality twice for the entropy dissipation terms in Lemma 1 to obtain

$$\int_{\Pi^d} \rho_t K * (\rho_t - \bar{\rho}_t) \cdot \nabla \log \frac{\rho_t}{\bar{\rho}_t} \leq \frac{\nu}{4} \int \rho_t |\nabla \log \frac{\rho_t}{\bar{\rho}_t}|^2 + \frac{1}{\nu} \int \rho_t |K * (\rho_t - \bar{\rho}_t)|^2,$$

and

$$\int_{\Pi^d} \rho_t \delta_t \cdot \nabla \log \frac{\rho_t}{\bar{\rho}_t} \leq \frac{\nu}{4} \int \rho_t |\nabla \log \frac{\rho_t}{\bar{\rho}_t}|^2 + \frac{1}{\nu} \int \rho_t |\delta_t|^2.$$

Furthermore,

$$\int \rho_t |K * (\rho_t - \bar{\rho}_t)|^2 \leq \|K\|_{L^\infty}^2 \|\rho_t - \bar{\rho}_t\|_{L^1}^2 \leq 2\|K\|_{L^\infty}^2 \int \rho_t \log \frac{\rho_t}{\bar{\rho}_t},$$

where the last inequality is simply the Csiszár–Kullback–Pinsker inequality [Villani et al., 2009]. Combining the above estimates, we obtain that given that $K \in L^\infty$,

$$\frac{\mathrm{d}}{\mathrm{d}t} \int_{\Pi^d} \rho_t \log \frac{\rho_t}{\bar{\rho}_t} = -\frac{\nu}{2} \int_{\Pi^d} \rho_t |\nabla \log \frac{\rho_t}{\bar{\rho}_t}|^2 + \frac{2\|K\|_{L^\infty}^2}{\nu} \int \rho_t \log \frac{\rho_t}{\bar{\rho}_t} + \frac{1}{\nu} \int \rho_t |\delta_t|^2.$$

Currently, we are not interested in the long time behavior, so we first ignore the negative term above to obtain that

$$\frac{\mathrm{d}}{\mathrm{d}t} \int_{\Pi^d} \rho_t \log \frac{\rho_t}{\bar{\rho}_t} \leq \frac{2\|K\|_{L^\infty}^2}{\nu} \int \rho_t \log \frac{\rho_t}{\bar{\rho}_t} + \frac{1}{\nu} \int \rho_t |\delta_t|^2.$$

By Gronwall inequality, we obtain that

$$\int_{\Pi^d} \rho_t \log \frac{\rho_t}{\bar{\rho}_t} \leq \frac{1}{\nu} \exp\Big(\frac{2\|K\|_{L^\infty}^2}{\nu} t\Big) \int_0^t \int \rho_s |\delta_s|^2 \mathrm{d}x\mathrm{d}s.$$

$\square$

## E.2    The McKean-Vlasov equation with Coulomb interactions

*Proof of Theorem 3.* We first prove the case when $\nu > 0$. Applying Cauchy-Schwarz inequality to the right-hand side of $\frac{\mathrm{d}}{\mathrm{d}t} E(\rho_t^f, \bar{\rho}_t)$ in Lemma 3, one has

$$\frac{\mathrm{d}}{\mathrm{d}t} E(\rho_t^f, \bar{\rho}_t) \leq \frac{1}{2} \int_{\mathcal{X}} \rho_t^f |\delta_t|^2 \mathrm{d}x$$
$$- \frac{1}{2} \int_{\mathcal{X}^2} K(x-y) \cdot \Big(\mathcal{A}[\bar{\rho}_t](x) - \mathcal{A}[\bar{\rho}_t](y)\Big) \mathrm{d}(\rho_t^f - \bar{\rho}_t)^{\otimes 2}(x,y).$$

By Lemma 5.2 in Bresch et al. [2019b], as long as the ground truth "velocity field" $\mathcal{A}[\bar{\rho}_t]$ is Lipschitz, i.e. $\mathcal{A}[\bar{\rho}] \in W^{1,\infty}$, or equivalently $\nabla^2 V \in W^{1,\infty}, \nabla^2 \log \bar{\rho}_t \in L^\infty, K * \bar{\rho}_t \in W^{1,\infty}$, using the particular structure introduced by the Coulomb interactions (note that $-\Delta g = \delta_0$ and $K = -\nabla g$), we have the estimate

$$- \frac{1}{2} \int_{\mathcal{X}^2} K(x-y) \cdot \Big(\mathcal{A}[\bar{\rho}_t](x) - \mathcal{A}[\bar{\rho}_t](y)\Big) \mathrm{d}(\rho_t^f - \bar{\rho}_t)^{\otimes 2}(x,y)$$
$$\leq C\|\nabla \mathcal{A}[\bar{\rho}_t]\|_{L^\infty} F(\rho_t^f, \bar{\rho}_t).$$

This estimate can be obtained either by Fourier method [Bresch et al., 2019b] or by the stress-energy tensor approach as in [Serfaty, 2020]. We emphasize that those assumptions made on $(\bar{\rho}_t)_{t\in[0,T]}$ can be obtained by propagating similar conditions on the initial data $\bar{\rho}_0$. This estimate actually holds for more general choices of $g$ or $K$. See more examples including Riesz kernels in [Bresch et al., 2019b]. Moreover, the Lipschitz regularity of $\mathcal{A}[\bar{\rho}_t]$ can also be relaxed a bit. See for instance in [Rosenzweig, 2022].

Combining previous two estimates, one has

$$\frac{\mathrm{d}}{\mathrm{d}t}E(\rho_t^f,\bar{\rho}_t) \le \frac{1}{2}\int_{\mathcal{X}}\rho_t^f\,|\delta_t|^2\mathrm{d}x + CC_1F(\rho_t^f,\bar{\rho}_t) \le \frac{1}{2}\int_{\mathcal{X}}\rho_t^f\,|\delta_t|^2\mathrm{d}x + CC_1E(\rho_t^f,\bar{\rho}_t).$$

Then applying Gronwall inequality concludes the proof of the case when $\nu > 0$.

Now we prove the deterministic case when $\nu = 0$. Now the relative entropy or KL divergence does not play a role since there is no Laplacian term in equation 2. Lemma 2 now reads

$$\frac{\mathrm{d}}{\mathrm{d}t}F(\rho_t^f,\bar{\rho}_t) = -\int_{\mathcal{X}}\rho_t^f|K*(\rho_t^f-\bar{\rho})|^2 - \int_{\mathcal{X}}\rho_t^f\,\delta_t\cdot K*(\rho_t^f-\bar{\rho}_t)$$
$$-\frac{1}{2}\int_{\mathcal{X}^2}K(x-y)\cdot\Big(\mathcal{A}[\bar{\rho}_t](x)-\mathcal{A}[\bar{\rho}_t](y)\Big)\mathrm{d}(\rho_t^f-\bar{\rho}_t)^{\otimes 2}(x,y).$$

Applying Cauchy-Schwarz to the 2nd term in the right-hand side above, we obtain that

$$\frac{\mathrm{d}}{\mathrm{d}t}F(\rho_t^f,\bar{\rho}_t) \le \frac{1}{2}\int_{\mathcal{X}}\rho_t^f\,|\delta_t|^2 - \frac{1}{2}\int_{\mathcal{X}^2}K(x-y)\cdot\Big(\mathcal{A}[\bar{\rho}_t](x)-\mathcal{A}[\bar{\rho}_t](y)\Big)\mathrm{d}(\rho_t^f-\bar{\rho}_t)^{\otimes 2}(x,y).$$

Again assuming that the "velocity field" $\mathcal{A}[\bar{\rho}_t](\cdot)$ is Lipschitz will give us

$$\frac{\mathrm{d}}{\mathrm{d}t}F(\rho_t^f,\bar{\rho}_t) \le \frac{1}{2}\int_{\mathcal{X}}\rho_t^f\,|\delta_t|^2 + CC_1F(\rho_t^f,\bar{\rho}_t).$$

Applying Gronwall inequality again conclude all the proof.

$\square$

# F Approximation Error of Neural Network

We show that in a function class $\mathcal{F}$ with sufficient capacity, there exists at least one element $\hat{f} \in \mathcal{F}$ such that $R(\hat{f})$ is small. In particular, we are interested in the function class of neural networks.

We will focus on the case where the domain is the torus $\mathcal{X} = \Pi^d$, i.e. a $d$ dimensional box with size $L$ endowed with the periodic boundary condition. For the simplicity of notations, we denote the underlying velocity by $\bar{f}_t = \mathcal{A}[\bar{\rho}_t]$, where the operator $\mathcal{A}$ is defined in equation 5.

In the following, we focus on the Coulomb case where $K$ is defined in equation 3. The Biot-Savart case (4) can be treated similarly.

*Proof of Theorem 4.* In equation 14, we showed that for any hypothesis velocity $f$, the EINN loss $R(f)$ admits the trajectory-wise reformulation:

$$R(f) = \int_{\mathcal{X}} d\boldsymbol{x}_0 \bar{\rho}_0(\boldsymbol{x}) \int_0^T dt \|\delta_t^f \circ X_t^f(\boldsymbol{x}_0)\|^2, \tag{59}$$

where we recall the definition of $\delta_t^f$ in equation 11. Note that, as a general principle, in this proof, we will use the superscript to emphasize the dependence on a velocity $f$, e.g. the flow map $X_t^f$.

From Assumption 2 we know that there exists $\hat{f} \in \mathcal{F}$ such that $\|\bar{f} - \hat{f}\|_{W^{2,\infty}(\mathcal{X})} \le \epsilon$. In the following, we show that $R(\hat{f})$ is small.

Define

$$A_{\boldsymbol{x}}^f(t) \overset{\text{def}}{=} \int_0^t \|\delta_s^f \circ X_s^f(\boldsymbol{x})\|^2 ds. \tag{60}$$

We have

$$R(f) = \int_{\mathcal{X}} A_{\boldsymbol{x}}^f(T) d\bar{\rho}_0(\boldsymbol{x}). \tag{61}$$

Recall that $\bar{f}$ denotes the underlying velocity and hence $\delta_t^{\bar{f}} \equiv 0$ and $\rho_t^{\bar{f}} \equiv \bar{\rho}_t$, where we recall that $\rho_t^{\bar{f}}$ is the solution to the continuity equation (7) with velocity field $\bar{f}$. We can bound

$$\frac{\partial}{\partial t} A_x^{\hat{f}}(t) = \|\delta_t^{\hat{f}} \circ X_t^{\hat{f}}(\boldsymbol{x})\|^2 = \|\delta_t^{\hat{f}} \circ X_t^{\hat{f}}(\boldsymbol{x}) - \delta_t^{\bar{f}} \circ X_t^{\bar{f}}(\boldsymbol{x})\|^2$$

$$\le 4\|\left(\hat{f}_t \circ X_t^{\hat{f}} - \bar{f}_t \circ X_t^{\bar{f}}\right)(\boldsymbol{x})\|^2 + 4\|\left(\nabla V \circ X_t^{\hat{f}} - \nabla V \circ X_t^{\bar{f}}\right)(\boldsymbol{x})\|^2$$

$$+ 4\|\left((K * \rho_t^{\hat{f}}) \circ X_t^{\hat{f}} - (K * \bar{\rho}_t) \circ X_t^{\bar{f}}\right)(\boldsymbol{x})\|^2 + 4\nu^2\|\left(\nabla \log \rho_t^{\hat{f}} \circ X_t^{\hat{f}} - \nabla \log \bar{\rho}_t \circ X_t^{\bar{f}}\right)(\boldsymbol{x})\|^2$$

$$= \text{①} + \text{②} + \text{③} + \text{④}. \tag{62}$$

We will bound each term on the R.H.S. individually. The following lemmas will be useful:

**Lemma 5.** *For two Lipschitz continuous velocity field $f_1, f_2 \in \mathcal{C}^1(\mathcal{X})$, we have for any $t \in [0, T]$*

$$\|X_t^{f_1}(\boldsymbol{x}) - X_t^{f_2}(\boldsymbol{x})\|^2 \le A_1(T)\|f_1 - f_2\|_{\mathcal{L}^\infty(\mathcal{X})}^2.$$

*Proof.* Denote $\boldsymbol{x}^i(t) = X_t^{f_i}(\boldsymbol{x}_0)$ for $i = 1, 2$.

$$\frac{d}{dt}\|\boldsymbol{x}^1(t) - \boldsymbol{x}^2(t)\|^2 \le \|\boldsymbol{x}^1(t) - \boldsymbol{x}^2(t)\|^2 + \|f_1(t, \boldsymbol{x}^1(t)) - f_2(t, \boldsymbol{x}^2(t))\|^2$$

$$\le C\|\boldsymbol{x}^1(t) - \boldsymbol{x}^2(t)\|^2 + \|f_1(t, \boldsymbol{x}^2(t)) - f_2(t, \boldsymbol{x}^2(t))\|^2$$

$$\le C\|\boldsymbol{x}^1(t) - \boldsymbol{x}^2(t)\|^2 + \|f_1 - f_2\|_{\mathcal{L}^\infty(\mathcal{X})}^2.$$

Using the Grönwall's inequality, we have the result. $\square$

**Lemma 6.** *Suppose that $f \in \mathcal{C}^1$ is Lipschitz continuous. We have that $X_t^f$ is an $A_2(T)$-Lipschitz continuous map. For $f \in \mathcal{L}^\infty(\mathcal{X})$, we have $\|X_t^f(\boldsymbol{x})\|^2 \le \|x\|^2 + t\|f\|_{\mathcal{L}^\infty}^2$.*

*Proof.* Denote $\boldsymbol{x}^i(t) = X_t^f(\boldsymbol{x}_0^i)$ for $i = 1, 2$.

$$\frac{\mathrm{d}}{\mathrm{d}t}\|\boldsymbol{x}^1(t) - \boldsymbol{x}^2(t)\|^2 \le \|\boldsymbol{x}^1(t) - \boldsymbol{x}^2(t)\|^2 + \|f(t, \boldsymbol{x}^1(t)) - f(t, \boldsymbol{x}^2(t))\|^2$$
$$\le C\|\boldsymbol{x}^1(t) - \boldsymbol{x}^2(t)\|^2.$$

Using the Grönwall's inequality, we have that $X_t^f$ is Lipschitz continuous. $\square$

**Lemma 7.** *For $f \in \mathcal{C}^2(\mathcal{X})$, suppose that $\nabla(\operatorname{div}f) \in \mathcal{L}^\infty(\mathcal{X})$ and $\mathcal{J}_f \in \mathcal{L}^\infty(\mathcal{X})$. Further suppose that the initial distribution $\bar\rho_0$ satisfies $\nabla \log \bar\rho_0 \in \mathcal{L}^\infty(\mathcal{X})$. We have that $\nabla \log \rho_t^f \circ X_t^f \in \mathcal{L}^\infty(\mathcal{X})$.*

*Proof.* Denote $\boldsymbol{x}(t) = X_t^f(\boldsymbol{x}_0)$. From equation 18, we have

$$\frac{\mathrm{d}}{\mathrm{d}t}\|\nabla \log \rho_t^f(\boldsymbol{x}(t))\|^2 \le C(1 + \|\nabla \log \rho_t^f(\boldsymbol{x}(t))\|^2). \tag{63}$$

Using the Grönwall's inequality, we have $\|\nabla \log \rho_t^f(\boldsymbol{x}(t))\|^2 < \infty$ for $\boldsymbol{x}_0 \in \mathcal{X}$. $\square$

**Bounding ① in equation 62**    We have

$$\|\left(\hat{f}_t \circ X_t^{\hat{f}} - \bar{f}_t \circ X_t^{\bar{f}}\right)(\boldsymbol{x})\|$$
$$\le \|\left(\hat{f}_t \circ X_t^{\hat{f}} - \bar{f}_t \circ X_t^{\hat{f}}\right)(\boldsymbol{x})\| + \|\left(\bar{f}_t \circ X_t^{\hat{f}} - \bar{f}_t \circ X_t^{\bar{f}}\right)(\boldsymbol{x})\|$$
$$\le \epsilon + \epsilon \cdot \|X_t^{\hat{f}}(\boldsymbol{x}) - X_t^{\bar{f}}(\boldsymbol{x})\| \le C_1(T)\epsilon.$$

**Bounding ② in equation 62**    We have from Assumption 2 and Lemma 5

$$\|\left(\nabla V \circ X_t^{\hat{f}} - \nabla V \circ X_t^{\bar{f}}\right)(\boldsymbol{x})\| \le C_2(T)\epsilon.$$

**Bounding ③ in equation 62**    Bounding ③ in equation 62 requires a more sophisticated analysis which is the major technical challenge of this proof. For the simplicity of notations, for a fixed $x$ and $y$, denote $\hat{\boldsymbol{x}}(t) = X_t^{\hat{f}}(\boldsymbol{x})$, $\bar{\boldsymbol{x}}(t) = X_t^{\bar{f}}(\boldsymbol{x})$, $\hat{\boldsymbol{y}}(t) = X_t^{\hat{f}}(\boldsymbol{y})$, $\bar{\boldsymbol{y}}(t) = X_t^{\bar{f}}(\boldsymbol{y})$. For any $\epsilon'$ which is to be determined later, we have

$$\|\left((K * \rho_t^{\hat{f}}) \circ X_t^{\hat{f}} - (K * \bar\rho_t) \circ X_t^{\bar{f}}\right)(\boldsymbol{x})\|^2 = \|\int_{\mathcal{X}} K(\hat{\boldsymbol{x}}(t) - \hat{\boldsymbol{y}}(t)) - K(\bar{\boldsymbol{x}}(t) - \bar{\boldsymbol{y}}(t))\mathrm{d}\bar\rho_0(\boldsymbol{y})\|^2$$

$$\le 2\|\int_{\|\hat{\boldsymbol{x}}(t) - \hat{\boldsymbol{y}}(t)\| \le \epsilon'} K(\hat{\boldsymbol{x}}(t) - \hat{\boldsymbol{y}}(t)) - K(\bar{\boldsymbol{x}}(t) - \bar{\boldsymbol{y}}(t))\mathrm{d}\bar\rho_0(\boldsymbol{y})\|^2 \tag{Ⓐ}$$

$$+ 2\|\int_{A_2(T)L \ge \|\hat{\boldsymbol{x}}(t) - \hat{\boldsymbol{y}}(t)\| \ge \epsilon'} K(\hat{\boldsymbol{x}}(t) - \hat{\boldsymbol{y}}(t)) - K(\bar{\boldsymbol{x}}(t) - \bar{\boldsymbol{y}}(t))\mathrm{d}\bar\rho_0(\boldsymbol{y})\|^2. \tag{Ⓑ}$$

Note that the upper bound on $\|\hat{\boldsymbol{x}}(t) - \hat{\boldsymbol{y}}(t)\|$ in Ⓑ comes from Lemma 6 and the facts that $\mathcal{X} = \Pi^d$ is bounded with size $L$. To bound Ⓐ, we have

$$\|\int_{\|\hat{\boldsymbol{x}}(t) - \hat{\boldsymbol{y}}(t)\| \le \epsilon'} K(\hat{\boldsymbol{x}}(t) - \hat{\boldsymbol{y}}(t)) - K(\bar{\boldsymbol{x}}(t) - \bar{\boldsymbol{y}}(t))\mathrm{d}\bar\rho_0(\boldsymbol{y})\|$$
$$\le \int_{\|\hat{\boldsymbol{x}}(t) - \hat{\boldsymbol{y}}(t)\| \le \epsilon'} \|K(\hat{\boldsymbol{x}}(t) - \hat{\boldsymbol{y}}(t))\| + \|K(\bar{\boldsymbol{x}}(t) - \bar{\boldsymbol{y}}(t))\|\mathrm{d}\bar\rho_0(\boldsymbol{y})$$
$$= \int_{\|\hat{\boldsymbol{x}}(t) - \hat{\boldsymbol{y}}(t)\| \le \epsilon'} \frac{1}{\|\hat{\boldsymbol{x}}(t) - \hat{\boldsymbol{y}}(t)\|^{d-1}} + \frac{1}{\|\bar{\boldsymbol{x}}(t) - \bar{\boldsymbol{y}}(t)\|^{d-1}}\mathrm{d}\bar\rho_0(\boldsymbol{y}) = Ⓒ + Ⓓ.$$

We can bound Ⓒ by

$$\int_{\|\hat{\boldsymbol{x}}(t) - \hat{\boldsymbol{y}}(t)\| \le \epsilon'} \frac{1}{\|\hat{\boldsymbol{x}}(t) - \hat{\boldsymbol{y}}(t)\|^{d-1}}\mathrm{d}\bar\rho_0(\boldsymbol{y}) = \int_{\|\hat{\boldsymbol{x}}(t) - y\| \le \epsilon'} \frac{1}{\|\hat{\boldsymbol{x}}(t) - y\|^{d-1}}\mathrm{d}\rho_t^{\hat{f}}(\boldsymbol{y}) \le \|\rho_t^{\hat{f}}\|_\infty \cdot \epsilon', \tag{64}$$

where in the above inequality we remove the singular term by using transforming to the polar coordinate system. To bound Ⓓ, we pick $\epsilon' = dA_1(T)\epsilon$, so that Lemma 5 implies

$$\{y \in \mathcal{X} | \|\hat{\boldsymbol{x}}(t) - \hat{\boldsymbol{y}}(t)\| \le \epsilon'\} \subseteq \{y \in \mathcal{X} | \|\bar{\boldsymbol{x}}(t) - \bar{\boldsymbol{y}}(t)\| \le \frac{d+2}{d}\epsilon'\},$$

and consequently

$$\int_{\|\hat{\boldsymbol{x}}(t)-\hat{\boldsymbol{y}}(t)\|\le\epsilon'}\frac{1}{\|\bar{\boldsymbol{x}}(t)-\bar{\boldsymbol{y}}(t)\|^{d-1}}\mathrm{d}\bar{\rho}_0(\boldsymbol{y})\le\int_{\|\bar{\boldsymbol{x}}(t)-\bar{\boldsymbol{y}}(t)\|\le2\epsilon'}\frac{1}{\|\bar{\boldsymbol{x}}(t)-\bar{\boldsymbol{y}}(t)\|^{d-1}}\mathrm{d}\bar{\rho}_0(\boldsymbol{y})\le\frac{d+2}{d}\|\rho_t^{\hat{f}}\|_\infty\cdot\epsilon'.\tag{65}$$

To bound Ⓑ, note that

$$\nabla K(\boldsymbol{x})=\frac{1}{\|x\|^{d+2}}\left(\|x\|^2 I-d\cdot x\otimes x\right)\Rightarrow\|\nabla K(\boldsymbol{x})\|\le\frac{d}{\|x\|^d}.\tag{66}$$

Denote $\boldsymbol{z}(t)=\min(\|\hat{\boldsymbol{x}}(t)-\hat{\boldsymbol{y}}(t)\|,\|\bar{\boldsymbol{x}}(t)-\bar{\boldsymbol{y}}(t)\|)$. Recall the choice of $\epsilon'=dA_1(T)\epsilon$. Using Lemma 5, we have that

$$\|\boldsymbol{z}(t)\|\ge\frac{d-2}{d}\|\hat{\boldsymbol{x}}(t)-\hat{\boldsymbol{y}}(t)\|.$$

Using the triangle inequality, we have

$$\left\|\int_{A_2(T)L\ge\|\hat{\boldsymbol{x}}(t)-\hat{\boldsymbol{y}}(t)\|\ge\epsilon'}K(\hat{\boldsymbol{x}}(t)-\hat{\boldsymbol{y}}(t))-K(\bar{\boldsymbol{x}}(t)-\bar{\boldsymbol{y}}(t))\mathrm{d}\bar{\rho}_0(\boldsymbol{y})\right\|$$

$$\le\int_{A_2(T)L\ge\|\hat{\boldsymbol{x}}(t)-\hat{\boldsymbol{y}}(t)\|\ge\epsilon'}\|K(\hat{\boldsymbol{x}}(t)-\hat{\boldsymbol{y}}(t))-K(\bar{\boldsymbol{x}}(t)-\bar{\boldsymbol{y}}(t))\|\mathrm{d}\bar{\rho}_0(\boldsymbol{y})$$

$$\le 2\epsilon'd\int_{A_2(T)L\ge\|\hat{\boldsymbol{x}}(t)-\hat{\boldsymbol{y}}(t)\|\ge\epsilon'}\frac{1}{\|\boldsymbol{z}(t)\|^d}\mathrm{d}\bar{\rho}_0(\boldsymbol{y})$$

$$\le 2\epsilon'd\int_{A_2(T)L\ge\|\hat{\boldsymbol{x}}(t)-\hat{\boldsymbol{y}}(t)\|\ge\epsilon'}(\frac{d}{d-2})^d\frac{1}{\|\hat{\boldsymbol{x}}(t)-\hat{\boldsymbol{y}}(t)\|^d}\mathrm{d}\bar{\rho}_0(\boldsymbol{y})$$

$$\le 2e\epsilon'd\|\rho_t^{\hat{f}}\|_\infty\int_{A_2(T)L\ge\|\hat{\boldsymbol{x}}(t)-y\|\ge.5\epsilon'}\frac{1}{\|y\|^d}\mathrm{d}y$$

$$=2e\epsilon'd\|\rho_t^{\hat{f}}\|_\infty\ln(A_2(T)L/\epsilon').$$

Combining the bounds of Ⓐ and Ⓑ, we have that

$$③\le C_3(T)(\epsilon\ln\frac{1}{\epsilon})^2.\tag{67}$$

**Bounding ④ in equation 62** Denote $\hat{\boldsymbol{x}}(t)=X_t^{\hat{f}}(\boldsymbol{x})$ and $\bar{\boldsymbol{x}}(t)=X_t^{\bar{f}}(\boldsymbol{x})$. Define

$$B_{\boldsymbol{x}}(t)\stackrel{\text{def}}{=}\left\|\left(\nabla\log\rho_t^{\hat{f}}\circ X_t^{\hat{f}}-\nabla\log\bar{\rho}_t\circ X_t^{\bar{f}}\right)(\boldsymbol{x})\right\|^2=\|\nabla\log\rho_t^{\hat{f}}(\hat{\boldsymbol{x}}(t))-\nabla\log\bar{\rho}_t(\bar{\boldsymbol{x}}(t))\|^2.\tag{68}$$

Computing its dynamics

$$\frac{\mathrm{d}}{\mathrm{d}t}B_{\boldsymbol{x}}(t)\le B_{\boldsymbol{x}}(t)+\left\|\frac{\mathrm{d}}{\mathrm{d}t}\left(\nabla\log\rho_t^{\hat{f}}(\hat{\boldsymbol{x}}(t))-\nabla\log\bar{\rho}_t(\bar{\boldsymbol{x}}(t))\right)\right\|^2\tag{69}$$

Recall equation 18. We have that

$$\frac{\mathrm{d}}{\mathrm{d}t}\nabla\log\rho_t^{\hat{f}}(\hat{\boldsymbol{x}}(t))=-\nabla\left(\nabla\cdot\hat{f}_t(\hat{\boldsymbol{x}}(t))\right)-\left(\mathcal{J}_{\hat{f}_t}(\hat{\boldsymbol{x}}(t))\right)^\top\nabla\log\rho_t^{\hat{f}}(\hat{\boldsymbol{x}}(t)),\tag{70}$$

$$\frac{\mathrm{d}}{\mathrm{d}t}\nabla\log\rho_t^{\bar{f}}(\bar{\boldsymbol{x}}(t))=-\nabla\left(\nabla\cdot\bar{f}_t(\bar{\boldsymbol{x}}(t))\right)-\left(\mathcal{J}_{\bar{f}_t}(\bar{\boldsymbol{x}}(t))\right)^\top\nabla\log\rho_t^{\bar{f}}(\bar{\boldsymbol{x}}(t)),\tag{71}$$

and hence

$$\left\|\frac{\mathrm{d}}{\mathrm{d}t}\left(\nabla\log\rho_t^{\hat{f}}(\hat{\boldsymbol{x}}(t))-\nabla\log\bar{\rho}_t(\hat{\boldsymbol{x}}(t))\right)\right\|^2$$

$$\le 2\|\nabla\left(\nabla\cdot\hat{f}_t(\hat{\boldsymbol{x}}(t))\right)-\nabla\left(\nabla\cdot\bar{f}_t(\bar{\boldsymbol{x}}(t))\right)\|^2+2\left\|\left(\mathcal{J}_{\hat{f}_t}(\hat{\boldsymbol{x}}(t))\right)^\top\nabla\log\rho_t^{\hat{f}}(\hat{\boldsymbol{x}}(t))-\left(\mathcal{J}_{\bar{f}_t}(\bar{\boldsymbol{x}}(t))\right)^\top\nabla\log\rho_t^{\bar{f}}(\bar{\boldsymbol{x}}(t))\right\|^2$$

$$=Ⓔ+Ⓕ.$$

We now bound these two terms individually. To bound Ⓔ,

$$\left\|\nabla\left(\nabla\cdot\hat{f}_t(\hat{\boldsymbol{x}}(t))\right)-\nabla\left(\nabla\cdot\bar{f}_t(\bar{\boldsymbol{x}}(t))\right)\right\|$$

$$\le\left\|\nabla\left(\nabla\cdot\hat{f}_t(\hat{\boldsymbol{x}}(t))\right)-\nabla\left(\nabla\cdot\hat{f}_t(\bar{\boldsymbol{x}}(t))\right)\right\|+\left\|\nabla\left(\nabla\cdot\hat{f}_t(\bar{\boldsymbol{x}}(t))\right)-\nabla\left(\nabla\cdot\bar{f}_t(\bar{\boldsymbol{x}}(t))\right)\right\|\le\epsilon+LA_1(T)\epsilon.$$

To bound Ⓕ

$$\|\left(\mathcal{J}_{\hat{f}_t}(\hat{\boldsymbol{x}}(t))\right)^\top \nabla \log \rho_t^{\hat{f}}(\hat{\boldsymbol{x}}(t)) - \left(\mathcal{J}_{\bar{f}_t}(\bar{\boldsymbol{x}}(t))\right)^\top \nabla \log \rho_t^{\bar{f}}(\bar{\boldsymbol{x}}(t))\|^2$$

$$\leq 2\|\left(\left(\mathcal{J}_{\hat{f}_t}(\hat{\boldsymbol{x}}(t))\right)^\top - \left(\mathcal{J}_{\bar{f}_t}(\bar{\boldsymbol{x}}(t))\right)^\top\right) \nabla \log \rho_t^{\hat{f}}(\hat{\boldsymbol{x}}(t))\|^2 + 2\|\left(\mathcal{J}_{\bar{f}_t}(\bar{\boldsymbol{x}}(t))\right)^\top \left(\nabla \log \rho_t^{\hat{f}}(\hat{\boldsymbol{x}}(t)) - \nabla \log \rho_t^{\bar{f}}(\bar{\boldsymbol{x}}(t))\right)\|^2$$

$$\leq 2(1 + LA_1(T))^2 \epsilon^2 + 2L^2 B_x(t).$$

Consequently, using Grönwall's inequality, we have that

$$④ \leq C_4(T)\epsilon^2. \tag{72}$$

Combining all the estimations for ① to ④, we have that

$$R(\hat{f}) \leq C(T)\epsilon^2 (\ln \frac{1}{\epsilon})^2, \tag{73}$$

for some constant $C(T)$ independent of $\epsilon$. $\qquad\square$

# G  Discussion on the Unbounded Case

In Section 3, we considered the torus case, i.e. $\mathcal{X}$ is a $d$-dimensional box with size $L$ with a periodic boundary condition. In this section, we consider the unbounded case, i.e. $\mathcal{X} = \mathbb{R}^d$. There are two major differences:

1. The first difference is that when $\mathcal{X} = \mathbb{R}^d$, we would obtain an additional integral-of-divergence term from the operation of integration by parts. When $\mathcal{X}$ is a torus, using Gauss's divergence theorem and the periodic boundary condition, this term immediately vanishes, which simplifies the analysis. In contrast, for the unbounded case, we need to handle this term by assuming some additional regularity conditions.

2. The second difference is that for the torus, it is reasonable to assume that the initial distribution $\bar{\rho}_0$ is fully supported, which is equivalent to the existence of some constant $c > 0$ such that $\bar{\rho}_0(\boldsymbol{x}) \geq c$ for all $\boldsymbol{x} \in \mathcal{X}$. Such an assumption will allow us to propagate the regularity of the initial distribution $\bar{\rho}_0$ to the solution at time $t$, i.e. $\bar{\rho}_t$. In contrast, for the unbounded case, such an assumption clearly does not hold since otherwise $\bar{\rho}_0$ would not be integrable. Consequently, we can no long propagate the regularity of the initial distribution and hence we need to directly make regularity assumptions on $\bar{\rho}_t$.

In the following, we will focus on addressing the first point and provide sufficient conditions such that Lemmas 1 and 2 can be recovered even in the unbounded case. To elaborate a bit on the second point, the theorems that are derived in the main body of the submission remain valid under the regularity assumptions given therein. However, unlike the torus case, it is difficult to establish these regularity results for the unbounded case by assuming the regularity of the initial distribution $\bar{\rho}_0$.

**Lemma 8** (Analogy of Lemma 1 in the unbounded case)**.** *Given the hypothesis velocity field* $f = f(t,x) \in C^1_{t,x}$. *Assume that* $(\rho^f_t)_{t \in [0,T]}$ *and* $(\bar{\rho}_t)_{t \in [0,T]}$ *are classical solutions to equation (7) and equation (6) respectively. It holds that (recall the definition of* $\delta_t$ *in equation (11))*

$$\frac{\mathrm{d}}{\mathrm{d}t} \int_{\mathcal{X}} \rho^f_t \log \frac{\rho^f_t}{\bar{\rho}_t} = -\nu \int_{\mathcal{X}} \rho^f_t |\nabla \log \frac{\rho^f_t}{\bar{\rho}_t}|^2 + \int_{\mathcal{X}} \rho^f_t K * (\rho^f_t - \bar{\rho}_t) \cdot \nabla \log \frac{\rho^f_t}{\bar{\rho}_t}$$
$$+ \int_{\mathcal{X}} \rho^f_t \delta_t \cdot \nabla \log \frac{\rho^f_t}{\bar{\rho}_t} - \int \mathrm{div}\Big(\rho^f_t \big(f_t \log \frac{\rho^f_t}{\bar{\rho}_t} - \bar{f}_t\big)\Big).$$

*where* $\mathcal{X}$ *is the tours* $\Pi^d$. *All the integrands are evaluated at* $\boldsymbol{x}$.

*Proof.* Recall the McKean-Vlasov equation 6 and the continuity equation 10. For simplicity, we write that $\rho_t = \rho^f_t$ and $\bar{f}_t = \mathcal{A}[\bar{\rho}_t]$. Then

$$\frac{\mathrm{d}}{\mathrm{d}t} \int \rho_t \log \frac{\rho_t}{\bar{\rho}_t} = -\int \mathrm{div}\big(\rho_t f_t\big) \log \frac{\rho_t}{\bar{\rho}_t} + \int \frac{\rho_t}{\bar{\rho}_t} \mathrm{div}\big(\bar{\rho}_t \bar{f}_t\big)$$
$$= \int \rho_t f_t \nabla \log \frac{\rho_t}{\bar{\rho}_t} - \int \nabla \frac{\rho_t}{\bar{\rho}_t} \bar{\rho}_t \bar{f}_t - \int \mathrm{div}\big(\rho_t (f_t \log \frac{\rho_t}{\bar{\rho}_t} - \bar{f}_t)\big).$$

We handle the first two terms on the R.H.S. just like the torus case and we can have the result.  $\square$

**Lemma 9** (Analogy of Lemma 2 in the unbounded case)**.** *Under the same assumptions as in Lemma 1, given the diffusion coefficient* $\nu \geq 0$, *it holds that (recall the definition of* $\delta_t$ *in equation (11))*

$$\frac{\mathrm{d}}{\mathrm{d}t} F(\rho^f_t, \bar{\rho}_t) = -\int_{\mathcal{X}} \rho^f_t \|K * (\rho^f_t - \bar{\rho}_t)\|^2 - \int_{\mathcal{X}} \rho^f_t \delta_t \cdot K * (\rho^f_t - \bar{\rho}_t) + \nu \int_{\mathcal{X}} \rho^f_t K * (\rho^f_t - \bar{\rho}_t) \cdot \nabla \log \frac{\rho^f_t}{\bar{\rho}_t}$$
$$- \frac{1}{2} \int_{\mathcal{X}^2} K(x-y) \cdot \Big(\mathcal{A}[\bar{\rho}_t](x) - \mathcal{A}[\bar{\rho}_t](y)\Big) \mathrm{d}(\rho^f_t - \bar{\rho}_t)^{\otimes 2}(x,y)$$
$$- \int \mathrm{div}\Big\{g * (\rho^f_t - \bar{\rho}_t)(x)(\rho^f_t(x) f_t(x) - \bar{\rho}_t(x) \bar{f}_t(x))\Big\} \mathrm{d}x$$

*where we recall that the operator* $\mathcal{A}$ *is defined in equation (5).*

*Proof.* Recall that $K = -\nabla g$. For simplicity, we write that $\rho_t = \rho_t^f$. Then

$$\frac{\mathrm{d}}{\mathrm{d}t}F(\rho_t, \bar{\rho}_t) = \frac{\mathrm{d}}{\mathrm{d}t}\frac{1}{2}\int_{\mathcal{X}^2} g(x-y)\mathrm{d}(\rho_t - \bar{\rho}_t)^{\otimes 2}(x,y)$$

$$= \int_{\mathcal{X}} g * (\rho_t - \bar{\rho}_t)(x)\big(\partial_t \rho_t(x) - \partial_t \bar{\rho}_t(x)\big)\mathrm{d}x$$

$$= -\int g * (\rho_t - \bar{\rho}_t)(x)\,\mathrm{div}\big\{\rho_t(x)f_t(x) - \bar{\rho}_t(x)\bar{f}_t(x)\big\}\mathrm{d}x$$

$$= \int \nabla g * (\rho_t - \bar{\rho}_t)(x)\big\{\rho_t(x)f_t(x) - \bar{\rho}_t(x)\bar{f}_t(x)\big\}\mathrm{d}x$$

$$\quad - \int \mathrm{div}\big\{g * (\rho_t - \bar{\rho}_t)(x)(\rho_t(x)f_t(x) - \bar{\rho}_t(x)\bar{f}_t(x))\big\}\mathrm{d}x$$

We handle the first term on the R.H.S. just like the torus case and we can have the result. $\qquad\square$

### G.1 Handling the Integral of the Divergence

Given a vector field, the following lemma provides a sufficient condition for the volume integral of its divergence over $\mathcal{X}$ to be zero. The idea is to construct a sequence of approximations to the integral of interest, each of which involves integration over a compact set. Consequently, Gauss's divergence theorem can be applied. We then utilize the dominant convergence theorem to exchange the order of the limit and integral.

**Lemma 10.** *For a vector function* $g : \mathbb{R}^d \to \mathbb{R}^d$ *which satisfies*

$$\int_{\mathbb{R}^d} \mathrm{d}\boldsymbol{x}\,|\mathrm{div}\,g(\boldsymbol{x})| < \infty \quad \text{and} \quad \int_{\mathbb{R}^d} \mathrm{d}\boldsymbol{x}\,\|g(\boldsymbol{x})\| < \infty, \tag{74}$$

*we have*

$$\int_{\mathbb{R}^d} \mathrm{d}\boldsymbol{x}\,\mathrm{div}\,g(\boldsymbol{x}) = 0. \tag{75}$$

*Proof.* Choose a cut-off function, indexed by $r > 1$, satisfying

$$\Phi_r(\boldsymbol{x}) = \begin{cases} 1, & \text{if} \quad \|\boldsymbol{x}\| \le r, \\ \frac{1}{2}(1 + \cos(\pi\|\boldsymbol{x}\|/r - 1)), & \text{if} \quad r < \|\boldsymbol{x}\| \le 2r, \\ 0, & \text{if} \quad 2r < \|\boldsymbol{x}\|. \end{cases} \tag{76}$$

We have $\|\nabla\Phi_r\|_{\mathcal{L}^\infty} = O(1/r)$. Using the chain rule of divergence, we have that

$$\mathrm{div}_{\boldsymbol{x}}(g \cdot \Phi_r) = \mathrm{div}_{\boldsymbol{x}}(g) \cdot \Phi_r + g \cdot \nabla\Phi_r. \tag{77}$$

We have $\int_{\mathbb{R}^d} \mathrm{d}\boldsymbol{x}\,\mathrm{div}_{\boldsymbol{x}}(g\Phi_r)(\boldsymbol{x}) = 0$ for all $r$ and $\boldsymbol{x}$, by noting $g\Phi_r(\boldsymbol{x}) = 0$ for $\|\boldsymbol{x}\| > 2r$ and using Gauss's divergence theorem on the $\boldsymbol{x}$ variable. Using conditions (74) and the dominated convergence theorem, we have

$$0 = \lim_{r\to\infty}\int_{\mathcal{X}} \mathrm{d}\boldsymbol{x}\,[\,\mathrm{div}_{\boldsymbol{x}}(g) \cdot \Phi_r](\boldsymbol{x}) + \lim_{r\to\infty}\int_{\mathcal{X}} \mathrm{d}\boldsymbol{x}\,[g \cdot \nabla\Phi_r](\boldsymbol{x})$$

$$= \int_{\mathcal{X}} \mathrm{d}\boldsymbol{x}\,\lim_{r\to\infty}[\,\mathrm{div}_{\boldsymbol{x}}(g) \cdot \Phi_r](\boldsymbol{x}) + \int_{\mathcal{X}} \mathrm{d}\boldsymbol{x}\,\lim_{r\to\infty}[g \cdot \nabla\Phi_r](\boldsymbol{x})$$

$$= \int_{\mathcal{X}} \mathrm{d}\boldsymbol{x}\,\mathrm{div}_{\boldsymbol{x}}(g)(\boldsymbol{x}),$$

where in the last equality, we use $g \cdot \nabla\Phi_r(\boldsymbol{x}) \le \|g(\boldsymbol{x})\|\|\nabla\Phi_r(\boldsymbol{x})\| \to 0$ as $r \to \infty$. $\qquad\square$

We now show that the divergence integrals in Lemmas 8 and 9 satisfy the requirements (74), under the following regularity assumptions on the hypothesis velocity field $f$, initial distribution $\bar{\rho}_0$, and the ground truth solution $\bar{\rho}$.

**Assumption 3.** *$f \in \mathrm{Lip}(\mathcal{X})$ and there exists some constant $L$, such that for all $t \in [0, T]$ and $\boldsymbol{x} \in \mathcal{X}$ $\|[\nabla(\mathrm{div}f)](t, \boldsymbol{x})\| \le L$.*

**Assumption 4.** *The initial distribution $\bar{\rho}_0$ satisfies*

$$\int_{\mathcal{X}} \mathrm{d}\boldsymbol{x}\,\bar{\rho}_0(\boldsymbol{x})(|\log\bar{\rho}_0(\boldsymbol{x})| + 1)(\|\boldsymbol{x}\| + 1)^{\alpha+1}(\|\nabla\log\bar{\rho}_0(\boldsymbol{x})\| + 1) < \infty \tag{78}$$

**Assumption 5.** *Suppose that the ground truth $\bar{\rho}_t \in \mathcal{L}^\infty$ is sufficiently regular such that*

$$|\log \bar{\rho}_t(\boldsymbol{x})| + \|\nabla \log \bar{\rho}_t(\boldsymbol{x})\| \le L(1 + \|\boldsymbol{x}\|)^\alpha \text{ and } \|\bar{f}_t(\boldsymbol{x})\| + |\operatorname{div} \bar{f}_t(\boldsymbol{x})| \le L(1 + \|\boldsymbol{x}\|)^\alpha \quad (79)$$

*holds for all $\boldsymbol{x} \in \mathcal{X}$ and $t \in [0, T]$ with some constant $\alpha$ and $L$. Here we denote $\bar{f}_t = \mathcal{A}[\rho_t]$.*

The following estimations of regularity will be helpful. The proof is deferred to the end of this section.

**Lemma 11.** *Under Assumption 3, we have the following estimations*

$$\|\boldsymbol{x}_t\|^2 \le \exp(t(1 + 2L^2))(\|\boldsymbol{x}_0\|^2 + 1)$$

$$|\log \rho_t^f(\boldsymbol{x}_t)| \le |\log \bar{\rho}_0(\boldsymbol{x}_0)| + Lt$$

$$\|\nabla \log \rho_t^f(\boldsymbol{x}_t)\|^2 \le \exp(t(1 + 2L^2))(\|\nabla \log \bar{\rho}_0(\boldsymbol{x}_0)\|^2 + 1).$$

We now show that the integrals of the divergence in Lemmas 12 and 13 are zero.

**Lemma 12.** *Under Assumptions 3 to 5, we have*

$$\int_{\mathcal{X}} \operatorname{div}\left(\rho_t(f_t \log \frac{\rho_t^f}{\bar{\rho}_t} - \bar{f}_t)\right) = 0. \quad (80)$$

*Proof.* To establish Lemma 12, we need to show that all the terms inside the divergence of equation 80 satisfy the integrability requirements (74) in Lemma 10, which are handled one by one in the following.

- We handle the term $\rho_t^f \log \rho_t^f f_t$.

  - To show that $\int_{\mathcal{X}} \mathrm{d}\boldsymbol{x} \, \|\rho_t^f \log \rho_t^f f_t(\boldsymbol{x})\| < \infty$

  $$\int_{\mathcal{X}} \mathrm{d}\boldsymbol{x} \, \|[\rho_t^f \log \rho_t^f f_t](\boldsymbol{x})\| = \int_{\mathcal{X}} \mathrm{d}\boldsymbol{x} \, \rho_t^f(\boldsymbol{x})\|[\log \rho_t^f f_t](\boldsymbol{x})\|$$

  $$= \int_{\mathcal{X}} \mathrm{d}\boldsymbol{x} \, \bar{\rho}_0(\boldsymbol{x}_0) \cdot |\log \rho_t^f(\boldsymbol{x}_t)| \cdot \|f_t(\boldsymbol{x}_t)\|$$

  $$\le \int_{\mathcal{X}} \mathrm{d}\boldsymbol{x} \, \bar{\rho}_0(\boldsymbol{x}_0) \cdot (|\log \bar{\rho}_0(\boldsymbol{x}_0)| + Lt) \cdot \exp(t(1 + 2L^2))(\|\boldsymbol{x}_0\| + 1) < \infty.$$

  - To show that $\int_{\mathcal{X}} \mathrm{d}\boldsymbol{x} \, |\operatorname{div}\left(\rho_t^f \log \rho_t^f f_t\right)(\boldsymbol{x})| < \infty$

  $$\operatorname{div}\left(\rho_t^f \log \rho_t^f f_t\right) = \operatorname{div} f_t \cdot \rho_t^f \log \rho_t^f + f_t \cdot \nabla(\rho_t^f \log \rho_t^f)$$

  $$= \operatorname{div} f_t \cdot \rho_t^f \cdot \log \rho_t^f + (f_t \cdot \nabla \rho_t^f) \cdot \log \rho_t^f + (f_t \cdot \nabla \log \rho_t^f) \cdot \rho_t^f$$

  $$= \rho_t^f \left(\operatorname{div} f_t \cdot \log \rho_t^f + (f_t \cdot \nabla \log \rho_t^f) \cdot (1 + \log \rho_t^f)\right)$$

  We now bound

  $$\int_{\mathcal{X}} \mathrm{d}\boldsymbol{x} \, \rho_t^f(\boldsymbol{x})|[\operatorname{div} f_t \cdot \log \rho_t^f](\boldsymbol{x})|$$

  $$= \int_{\mathcal{X}} \mathrm{d}\boldsymbol{x} \, \bar{\rho}_0(\boldsymbol{x}_0)|[\operatorname{div} f_t \cdot \log \rho_t^f](\boldsymbol{x}_t)| \le \int_{\mathcal{X}} \mathrm{d}\boldsymbol{x} \, \bar{\rho}_0(\boldsymbol{x}_0) \cdot L \cdot (tL + |\log \bar{\rho}_0(\boldsymbol{x}_0)|) < \infty.$$

  and

  $$\int_{\mathcal{X}} \mathrm{d}\boldsymbol{x} \, \rho_t^f(\boldsymbol{x})|[(f_t \cdot \nabla \log \rho_t^f) \cdot (1 + \log \rho_t^f)](\boldsymbol{x})|$$

  $$= \int_{\mathcal{X}} \mathrm{d}\boldsymbol{x} \, \bar{\rho}_0(\boldsymbol{x}_0)|[(f_t \cdot \nabla \log \rho_t^f) \cdot (1 + \log \rho_t^f)](\boldsymbol{x}_t)|$$

  $$\le \int_{\mathcal{X}} \mathrm{d}\boldsymbol{x} \, \bar{\rho}_0(\boldsymbol{x}_0)L(1 + \|\boldsymbol{x}_0\|)\exp(t(1 + 2L^2))(\|\nabla \log \bar{\rho}_0(\boldsymbol{x}_0)\| + 1)(1 + Lt + |\log \bar{\rho}_0(\boldsymbol{x}_0)|) < \infty.$$

- We handle the term $\rho_t^f \log \bar{\rho}_t f_t$.

- To show that $\int_{\mathcal{X}} d\boldsymbol{x} \, \| \rho_t^f \log \bar{\rho}_t f_t(\boldsymbol{x}) \| < \infty$

$$\int_{\mathcal{X}} d\boldsymbol{x} \, \rho_t^f \| \log \bar{\rho}_t f_t(\boldsymbol{x}) \| = \int_{\mathcal{X}} d\boldsymbol{x} \, \bar{\rho}_0 \| [\log \bar{\rho}_t f_t](\boldsymbol{x}_t) \|$$

$$\leq \int_{\mathcal{X}} d\boldsymbol{x} \, \bar{\rho}_0(\boldsymbol{x}_0) |\log \bar{\rho}_t(\boldsymbol{x}_t)| \| f_t(\boldsymbol{x}_t) \| \leq \int_{\mathcal{X}} d\boldsymbol{x} \, \bar{\rho}_0(\boldsymbol{x}_0) L^2 (1 + \|\boldsymbol{x}_t\|)^{\alpha+1} < \infty.$$

- To show that $\int_{\mathcal{X}} d\boldsymbol{x} \, | \mathrm{div}\left( \rho_t^f \log \bar{\rho}_t f_t \right)(\boldsymbol{x})| < \infty$

$$\mathrm{div}\left( \rho_t^f \log \bar{\rho}_t f_t \right) = \mathrm{div} f_t \cdot \rho_t^f \log \bar{\rho}_t + f_t \cdot \nabla (\rho_t^f \log \rho_t)$$

$$= \mathrm{div} f_t \cdot \rho_t^f \cdot \log \bar{\rho}_t + (f_t \cdot \nabla \rho_t^f) \cdot \log \bar{\rho}_t + (f_t \cdot \nabla \log \rho_t) \cdot \rho_t^f$$

$$= \rho_t^f \left( \mathrm{div} f_t \cdot \log \bar{\rho}_t + (f_t \cdot \nabla \log \rho_t^f) \log \bar{\rho}_t + f_t \cdot \nabla \log \rho_t \right)$$

We now bound

$$\int_{\mathcal{X}} d\boldsymbol{x} \, \rho_t^f(\boldsymbol{x}) |\mathrm{div} f_t(\boldsymbol{x})| \cdot |\log \bar{\rho}_t(\boldsymbol{x})| = \int_{\mathcal{X}} d\boldsymbol{x} \, \bar{\rho}_0(\boldsymbol{x}_0) |\mathrm{div} f_t(\boldsymbol{x}_t)| \cdot |\log \bar{\rho}_t(\boldsymbol{x}_t)|$$

$$\leq \int_{\mathcal{X}} d\boldsymbol{x} \, \bar{\rho}_0(\boldsymbol{x}_0) L(1 + \|\boldsymbol{x}_t\|)(1 + \|\boldsymbol{x}_t\|)^{\alpha} < \infty.$$

$$\int_{\mathcal{X}} d\boldsymbol{x} \, \rho_t^f(\boldsymbol{x}) |[(f_t \cdot \nabla \log \rho_t^f) \log \rho_t](\boldsymbol{x})| = \int_{\mathcal{X}} d\boldsymbol{x} \, \bar{\rho}_0(\boldsymbol{x}_0) |[(f_t \cdot \nabla \log \rho_t^f) \log \rho_t](\boldsymbol{x}_t)|$$

$$\leq \int_{\mathcal{X}} d\boldsymbol{x} \, \bar{\rho}_0(\boldsymbol{x}_0) \| f_t(\boldsymbol{x}_t) \| \| \nabla \log \rho_t^f(\boldsymbol{x}_t) \| |\log \bar{\rho}_t(\boldsymbol{x}_t)|$$

$$\leq \int_{\mathcal{X}} d\boldsymbol{x} \, \exp(t(1 + 2L^2))(\| \nabla \log \bar{\rho}_0(\boldsymbol{x}_0) \| + 1) L(1 + \|\boldsymbol{x}_t\|)(1 + \|\boldsymbol{x}_t\|)^{\alpha} < \infty.$$

$$\int_{\mathcal{X}} d\boldsymbol{x} \, \rho_t^f(\boldsymbol{x}) |[f_t \cdot \nabla \log \rho_t](\boldsymbol{x})| = \int_{\mathcal{X}} d\boldsymbol{x} \, \bar{\rho}_0(\boldsymbol{x}_0) \| f_t(\boldsymbol{x}_t) \| \| \nabla \log \bar{\rho}_t(\boldsymbol{x}_t) \|$$

$$\leq \int_{\mathcal{X}} d\boldsymbol{x} \, \bar{\rho}_0(\boldsymbol{x}_0) \| f_t(\boldsymbol{x}_t) \| \| \nabla \log \bar{\rho}_t(\boldsymbol{x}_t) \| \leq \int_{\mathcal{X}} d\boldsymbol{x} \, \bar{\rho}_0(\boldsymbol{x}_0) L(1 + \|\boldsymbol{x}_t\|)(1 + \|\boldsymbol{x}_t\|)^{\alpha} < \infty.$$

- We handle the term $\rho_t^f \bar{f}_t$.

  - To show that $\int_{\mathcal{X}} d\boldsymbol{x} \, \| [\rho_t^f \bar{f}_t](\boldsymbol{x}) \| < \infty$

$$\int_{\mathcal{X}} d\boldsymbol{x} \, \rho_t^f(\boldsymbol{x}) \| \bar{f}_t(\boldsymbol{x}) \| = \int_{\mathcal{X}} d\boldsymbol{x} \, \bar{\rho}_0(\boldsymbol{x}_0) \| \bar{f}_t(\boldsymbol{x}_t) \| < \infty$$

  - To show that $\int_{\mathcal{X}} d\boldsymbol{x} \, | \mathrm{div}(\rho_t^f \bar{f}_t)(\boldsymbol{x}) | < \infty$

$$\int_{\mathcal{X}} \mathrm{div}(\rho_t^f \bar{f}_t) = \int_{\mathcal{X}} \nabla \rho_t^f \cdot \bar{f}_t + \rho_t^f \, \mathrm{div} \bar{f}_t = \int_{\mathcal{X}} \rho_t^f \left( \nabla \log \rho_t^f \cdot \bar{f}_t + \mathrm{div} \bar{f}_t \right)$$

$$= \int_{\mathcal{X}} d\boldsymbol{x}_0 \, \bar{\rho}_0(\boldsymbol{x}_0) \left( \nabla \log \rho_t^f(\boldsymbol{x}_t) \cdot \bar{f}_t(\boldsymbol{x}_t) + \mathrm{div} \bar{f}_t(\boldsymbol{x}_t) \right) < \infty,$$

using the polynomial growth assumption on the ground truth velocity field $\bar{f}_t$.

$\square$

We now focus on addressing the integral-of-divergence term in Lemma 9. The following result will be useful.

**Remark 2.** *Let $X_t^f$ be the flow map generated by the velocity field $f \in \mathrm{Lip}(\mathbb{R}^d)$. We have that $X_t^f \in \mathrm{Lip}(\mathbb{R}^d)$ and that $\rho_t^f = X_t^f \sharp \bar{\rho}_0$ remains bounded for $t \in [0, T]$ if $\bar{\rho}_0$ is bounded on $\mathbb{R}^d$. This can be established using the change-of-variable formula of the probability density function.*

**Lemma 13.** *Under Assumptions 3 to 5, we have*

$$\int_{\mathcal{X}} d\boldsymbol{x} \, \mathrm{div}\left\{ g * (\rho_t^f - \bar{\rho}_t)(\boldsymbol{x})(\rho_t^f(\boldsymbol{x}) f_t(\boldsymbol{x}) - \bar{\rho}_t(\boldsymbol{x}) \bar{f}_t(\boldsymbol{x})) \right\} = 0. \tag{81}$$

*Proof.* Denote $h = g * (\rho_t - \bar{\rho}_t)(\boldsymbol{x})(\rho_t(\boldsymbol{x})f_t(\boldsymbol{x}) - \bar{\rho}_t(\boldsymbol{x})\bar{f}_t(\boldsymbol{x}))$. To show that $h \in \mathcal{L}^1(\mathcal{X})$, we can show that, after splitting into simple terms, every term from $h$ is in $\mathcal{L}^1$. In the following, we show

$$g * \rho_t(\boldsymbol{x})\rho_t(\boldsymbol{x})f_t(\boldsymbol{x}) \in \mathcal{L}^1(\mathcal{X}).$$

Other terms can be proved similarly. First, we show that $g * \rho_t \in \mathcal{L}^\infty(\mathcal{X})$ if $\rho_t \in \mathcal{L}^\infty(\mathcal{X})$. For any constant $C$, we have

$$\begin{aligned}
g * \rho_t(\boldsymbol{x}) &= \int_\mathcal{X} g(\boldsymbol{x} - \boldsymbol{y})\rho_t(\boldsymbol{y})\mathrm{d}\boldsymbol{y} \\
&= \int_{\|\boldsymbol{x}-\boldsymbol{y}\|\le C} g(\boldsymbol{x} - \boldsymbol{y})\rho_t(\boldsymbol{y})\mathrm{d}\boldsymbol{y} + \int_{\|\boldsymbol{x}-\boldsymbol{y}\|>C} g(\boldsymbol{x} - \boldsymbol{y})\rho_t(\boldsymbol{y})\mathrm{d}\boldsymbol{y} \\
&\le \|\rho_t\|_{\mathcal{L}^\infty(\mathcal{X})} \int_{\|\boldsymbol{x}-\boldsymbol{y}\|\le C} \|\boldsymbol{x} - \boldsymbol{y}\|^{2-d}\mathrm{d}\boldsymbol{y} + C^{2-d} \int_{\|\boldsymbol{x}-\boldsymbol{y}\|>C} \rho_t(\boldsymbol{y})\mathrm{d}\boldsymbol{y} \\
&\le \|\rho_t\|_{\mathcal{L}^\infty(\mathcal{X})}C^2 + C^{2-d},
\end{aligned}$$

where in the last inequality, we use

$$\int_{\|\boldsymbol{x}-\boldsymbol{y}\|\le C} \|\boldsymbol{x} - \boldsymbol{y}\|^{2-d}\mathrm{d}\boldsymbol{y} = \int_{\|\boldsymbol{y}\|\le C} \|\boldsymbol{y}\|^{2-d}\mathrm{d}\boldsymbol{y} \le \int_{0\le r\le C} r^{2-d}\mathrm{d}r \int J_\theta \mathrm{d}\theta \le \int_{0\le r\le C} \mathrm{d}r\, r \le C^2.$$

Here $J_\theta$ denotes the determinant of the Jacobian obtained from changing to the polar coordinate, which is bounded by $r^{d-1}$. We hence obtain

$$\int_\mathcal{X} \mathrm{d}\boldsymbol{x}\, \|g * \rho_t(\boldsymbol{x})\rho_t(\boldsymbol{x})f_t(\boldsymbol{x})\| \le C' \int_\mathcal{X} \mathrm{d}\boldsymbol{x}\, \rho_t(\boldsymbol{x})\|f_t(\boldsymbol{x})\| = C' \int_\mathcal{X} \mathrm{d}\boldsymbol{x}_0\, \bar{\rho}_0(\boldsymbol{x}_0)\|f_t(\boldsymbol{x}_t)\| < \infty,$$

where we use $f_t \in \mathrm{Lip}(\mathcal{X})$ and the estimation in Lemma 11.

Similarly, to show that $\mathrm{div}(h) \in \mathcal{L}^1(\mathcal{X})$, we can show that, after splitting into simple terms, every term from $\mathrm{div}(h)$ is in $\mathcal{L}^1$. In the following, we show that

$$\nabla g * \rho_t(\boldsymbol{x})\rho_t(\boldsymbol{x})f_t(\boldsymbol{x}) \in \mathcal{L}^1(\mathcal{X}) \text{ and } g * \rho_t(\boldsymbol{x})\nabla\rho_t(\boldsymbol{x}) \cdot f_t(\boldsymbol{x}) \in \mathcal{L}^1(\mathcal{X}).$$

Other terms can be proved similarly.

To show that $\nabla g * \rho_t(\boldsymbol{x})\rho_t(\boldsymbol{x})f_t(\boldsymbol{x}) \in \mathcal{L}^1(\mathcal{X})$, we first show that $\nabla g * \rho_t(\boldsymbol{x}) \in \mathcal{L}^\infty(\mathcal{X})$ for $\rho_t \in \mathcal{L}^\infty(\mathcal{X})$. We can then apply the same argument as above to establish the absolute integrability of the whole term.

$$\begin{aligned}
\|\nabla g * \rho_t(\boldsymbol{x})\| &\le \int_\mathcal{X} \|\nabla g(\boldsymbol{x} - \boldsymbol{y})\|\rho_t(\boldsymbol{y})\mathrm{d}\boldsymbol{y} \\
&= \int_{\|\boldsymbol{x}-\boldsymbol{y}\|\le C} \|\nabla g(\boldsymbol{x} - \boldsymbol{y})\|\rho_t(\boldsymbol{y})\mathrm{d}\boldsymbol{y} + \int_{\|\boldsymbol{x}-\boldsymbol{y}\|>C} \|\nabla g(\boldsymbol{x} - \boldsymbol{y})\|\rho_t(\boldsymbol{y})\mathrm{d}\boldsymbol{y} \\
&\le \|\rho_t\|_{\mathcal{L}^\infty(\mathcal{X})} \int_{\|\boldsymbol{x}-\boldsymbol{y}\|\le C} \|\boldsymbol{x} - \boldsymbol{y}\|^{1-d}\mathrm{d}\boldsymbol{y} + C^{1-d} \int_{\|\boldsymbol{x}-\boldsymbol{y}\|>C} \rho_t(\boldsymbol{y})\mathrm{d}\boldsymbol{y} \\
&\le \|\rho_t\|_{\mathcal{L}^\infty(\mathcal{X})}C + C^{1-d}.
\end{aligned}$$

To show that $g * \rho_t(\boldsymbol{x})\nabla\rho_t(\boldsymbol{x}) \cdot f_t(\boldsymbol{x}) \in \mathcal{L}^1(\mathcal{X})$, we use the fact that $g * \rho_t \in \mathcal{L}^\infty(\mathcal{X})$ and that

$$\int_\mathcal{X} \mathrm{d}\boldsymbol{x}\, \nabla\rho_t(\boldsymbol{x}) \cdot f_t(\boldsymbol{x}) = \int_\mathcal{X} \mathrm{d}\boldsymbol{x}_0\, \bar{\rho}_0(\boldsymbol{x}_0)\nabla\log\rho_t(\boldsymbol{x}_t) \cdot f_t(\boldsymbol{x}_t). \tag{82}$$

Using the estimation in Lemma 11 and that $f_t \in \mathrm{Lip}(\mathcal{X})$, we obtain the result. $\square$

*Proof of Lemma 11.*

$$\frac{\mathrm{d}}{\mathrm{d}t}\|\boldsymbol{x}_t\|^2 \le \|\boldsymbol{x}_t\|^2 + \|\bar{f}_t(\boldsymbol{x}_t)\|^2 \le \|\boldsymbol{x}_t\|^2 + 2L^2(1 + \|\boldsymbol{x}_t\|^2) = (1 + 2L^2)\|\boldsymbol{x}_t\|^2 + 2L^2. \tag{83}$$

Using Grönwall's inequality, we have

$$\|\boldsymbol{x}_t\|^2 \le \exp(t(1 + 2L^2))(\|\boldsymbol{x}_0\|^2 + 2L^2/(1 + 2L^2)) \le \exp(t(1 + 2L^2))(\|\boldsymbol{x}_0\|^2 + 1). \tag{84}$$

We have

$$\frac{\mathrm{d}}{\mathrm{d}t}\log\rho_t^f(\boldsymbol{x}_t) = -\mathrm{div}\bar{f}_t(\boldsymbol{x}_t) \tag{85}$$

We have

$$\frac{\mathrm{d}}{\mathrm{d}t}\nabla\log\rho_t^f(\boldsymbol{x}_t) = -\nabla\left(\operatorname{div}\bar{f}_t(\boldsymbol{x}_t)\right) - \left(\mathcal{J}_{\bar{f}_t}(\boldsymbol{x}_t)\right)^\top \nabla\log\rho_t^f(\boldsymbol{x}_t) \tag{86}$$

$$\frac{\mathrm{d}}{\mathrm{d}t}\|\nabla\log\rho_t^f(\boldsymbol{x}_t)\|^2 \le \|\nabla\log\rho_t^f(\boldsymbol{x}_t)\|^2 + 2\|\nabla\left(\operatorname{div}\bar{f}_t(\boldsymbol{x}_t)\right)\|^2 + 2\|\left(\mathcal{J}_{\bar{f}_t}(\boldsymbol{x}_t)\right)^\top \nabla\log\rho_t^f(\boldsymbol{x}_t)\|^2 \tag{87}$$

$$\le \|\nabla\log\rho_t^f(\boldsymbol{x}_t)\|^2(1+2\|\mathcal{J}_{\bar{f}_t}(\boldsymbol{x}_t)\|^2) + 2\|\nabla\left(\operatorname{div}\bar{f}_t(\boldsymbol{x}_t)\right)\|^2 \tag{88}$$

$$\le \|\nabla\log\rho_t^f(\boldsymbol{x}_t)\|^2(1+2L^2) + 2L^2 \tag{89}$$

Using Grönwall's inequality, we have

$$\|\nabla\log\rho_t^f(\boldsymbol{x}_t)\|^2 \le \exp(t(1+2L^2))(\|\nabla\log\bar{\rho}_0(\boldsymbol{x}_0)\|^2 + 1). \tag{90}$$

$\square$