# OpenReview forum: "Entropy-dissipation Informed Neural Network for McKean-Vlasov Type PDEs"
_NeurIPS.cc/2023/Conference — NeurIPS 2023 poster_

### Official Review · Reviewer_A2vP · 2023-06-29

**Soundness:** 3 good
**Presentation:** 3 good
**Contribution:** 3 good
**Rating:** 7
**Confidence:** 4

**Summary:**

This work focuses on deriving a uniform-in-time bound to the KL divergence between the NN-derived solution $\rho_t^f$ and the true solution $\bar{\rho}_t$, where $\rho_t^f$ is obtained via a push forward of samples of $\bar{\rho}_0$ with velocity parameterized by the neural network $f$. The neural network $f$ is trained to match the operator $\mathcal{A}[\rho]$ via the the Entropy-dissipation Informed Neural Network (EINN) loss given as an expectation over $\bar{\rho}_0$ of the integral from time $0$ to $T$ of the difference between $f$ and $\mathcal{A}[\rho^f]$. The method is then demonstrated to perform better than SOTA on McKean-Vlasov equations with singular interaction kernels: Coulomb interaction and Biot-Savart interaction.

**Strengths:**

1. Uniform-in-time bound to the KL divergence for finite duration $T$.
2. Method was shown to work on singular interaction kernels: Coulomb interaction,\ and Biot-Savart interaction.
3. Singularity for Biot-Savart kernel was removed in gradient computation during training.

**Weaknesses:**

1. Addressed in the last paragraph of section 4, the similarity in methodology to Shen et al. [2022]. Though the implementation is similar, the theoretical contribution is different. Perhaps include a comparison of their result in Figure 1 with and without generalization to MVE. And perhaps discuss the generalization to MVE.
2. Also addressed in the second last paragraph of section 4, the similar in loss function to Random Deep Vortex Network (RDVN) of Zhang et al. [2022]. Though the losses are similar, the authors mention a lack of discrepancy control in Zhang et al. [2022]. Perhaps provide more discussion.


**Questions:**

1. Maybe I am misunderstanding, how is using a NN to parameterize $\mathcal{A}[\rho^f]$ helping in computation? $\rho_t$ is still computed via push forward of $\rho_0$. Would using the true $\mathcal{A}[\rho]$ be better? Or is the goal to first compute $f$ as a surrogate to $\mathcal{A}[\rho]$ then compute $\rho^f$ from the PDE with $f$ as a surrogate for $\mathcal{A}[\rho]$?
2. Why is $K_c$ set to $0$ instead of clipping some large value (line 173, equation (19))?


**Limitations:**

1. Addressed in Section 3, approximation error of neural network that if a neural network can represent the true $\mathcal{A}[\rho]$, EINN loss is zero. Perhaps reference some papers on neural networks designed to represent $f$ for MVE.

---

> ### Author Rebuttal · Authors · 2023-08-06
>
> We thank the reviewer for making valuable comments. In the following, we address them one by one. Our response will be incorporated in our revision.
>
> > Addressed in the last paragraph of section 4, the similarity in methodology to Shen et al. [2022] ... And perhaps discuss the generalization to MVE.
>
> As acknowledged by the reviewer, EINN can handle the McKean-Vlasov equation which is more general than the Fokker-Planck equation considered in Shen et al. [2022]. For the purpose of comparison, let us now focus on the Fokker-Planck equation. In this special instance, the difference between EINN and Shen et al. [2022] is that the former uses functional $\mathcal{L}^2$ norm of the perturbation $\delta^f$ while the latter requires the Sobolev norm, making their method much more complicated to implement. We believe the additional terms of the higher order derivatives in the Sobolev norm are artifacts for establishing the theoretical guarantee therein. In fact, with our theoretical analysis, we can directly recover a stronger guarantee (convergence in terms of KL-divergence) than that of Shen et al. [2022] (convergence in terms of Wasserstein-2 distance), by setting $K=0$ (please see Theorem 6 in the appendix).
>
> In this regard, EINN can be regarded as an improvement (and simplification) of Shen et al. [2022] for the Fokker-Planck equation and a generalization of their work to the MVE case.
>
> In terms of directly generalizing the Sobolev norm loss of Shen et al. [2022] to the MVE, we are not sure how the higher order gradient of the perturbation $\delta^f$ can be computed (or even defined) for the (possibly singular) interaction term. To be specific, in the instance of MVE with Coulomb interaction, we have that $K(x) = c \frac{x}{\|x\|^d}$ for $d\geq 2$ where $c$ is some constant and $d$ is the ambient dimension. The Sobolev norm of $\delta^f$ involves the gradient of $K(x)$, which contains terms of the order $ \frac{1}{\|x\|^d}$. This term is no longer locally integrable, i.e. for any $\epsilon > 0$, $\int_{\|x\|\leq\epsilon} \frac{1}{\|x\|^d} \mathrm{d} x = \infty$. We will discuss this in our revision.
>
> > Also addressed in the second last paragraph of section 4, the similar in loss function to Random Deep Vortex Network (RDVN) of Zhang et al. [2022] ... Perhaps provide more discussion.
>
> While both RDVN and EINN share a similar form of minimizing functional $\mathcal{L}^2$ norm of some perturbation function, we emphasize that perturbation functions therein are of different meanings: The perturbation function of RDVN, represented by $u_t^\theta - K \ast \rho_t^\theta$, captures the error in estimating the interaction term in the 2D-NSE and the diffusion term ($\Delta \rho$ in the MVE in Eq.2) is incorporated in the SDE (see line 313). In contrast, the perturbation function of EINN (defined in Eq.11) is derived from the sensitive analysis of some Lyapunov function and should be understood as the error of estimating the **entire velocity field** of the 2D-NSE. Importantly, the diffusion term is also included in the EINN loss. Consequently, the EINN loss captures the **full information** of the velocity of the 2D-NSE while the RDVN only captures a part of it (the interaction part). We believe this is the reason why the authors of RDVN did not provide rigorous solution recover guarantee in the work Zhang et al. [2022].
>
> > Maybe I am misunderstanding how is using a NN to parameterize $\mathcal{A}[\rho^f]$ helping in computation? Would using the true  $\mathcal{A}[\rho]$ be better?
>
> The EINN objective (Eq.8) is derived from the sensitivity analysis, as discussed in section 2 of the paper. One of the practical benefits of using $\mathcal{A}[\rho^f]$ in the EINN objective is that $\mathcal{A}[\rho^f]\circ X_t$ can be computed using **only the initial distribution $\rho_0$ and the hypothesis velocity field $f$** (please see a detailed discussion in section 2.1 in our paper). It is possible that replacing $\mathcal{A}[\rho^f]$ with the true velocity field $\mathcal{A}[\rho]$ in the EINN loss would lead to a similar theoretical guarantee, but since $\mathcal{A}[\rho]$ is **unknown** (in fact this is the quantity we want to learn in the first place), the objective function after this replacement does not yield a practical algorithm.
>
> > Or is the goal to first compute $f$ as a surrogate to $\mathcal{A}[\rho]$ then compute $\rho^f$ from the PDE with $f$ as a surrogate for $\mathcal{A}[\rho]$?
>
> Yes, the reviewer is correct.
>
> > Why is $K_c$ set to $0$ instead of clipping some large value (line 173, equation (19))?
>
> We pick $K_c = 0$ for the sake of simplicity in our discussion, a choice that is also common practice in the literature on the mean-field limit. However, we also point out that the analysis below line 173 remains the same if we pick $K_c$ to be any function with the level of singularity less than $\frac{1}{\|x\|^{d-1}}$. Here by "same", we mean that the dependence of the cut-off error on the threshold $c$ remains $O(c)$ in Eq.20, when the cut-off threshold is small ($c\rightarrow 0$). Consequently, one can try larger values for $K_c$ with $\|x\|\leq c$, which might leads to better practical performance.
>
> > Addressed in Section 3, approximation error of neural network that if a neural network can represent the true $\mathcal{A}[\rho]$, EINN loss is zero. Perhaps reference some papers on neural networks designed to represent $f$ for MVE.
>
> One of the neural network structures that meets the requirement in Assumption 2 is [T. De Ryck, S. Lanthaler, and S. Mishra. On the approximation of functions by tanh neural net- works. Neural Networks, 143:732–750, 2021](https://www.sciencedirect.com/science/article/pii/S0893608021003208), as discussed in line 274 of our paper.

---

> > ### Comment · Reviewer_A2vP · 2023-08-15
> >
> > I would like to thank the authors for carefully answering my many questions. I am positive towards this paper and am raising my score to 7. I have read all the other reviews and rebuttals.

---

### Official Review · Reviewer_Jhg5 · 2023-07-04

**Soundness:** 3 good
**Presentation:** 2 fair
**Contribution:** 3 good
**Rating:** 5
**Confidence:** 3

**Summary:**

This paper proposes a new method for learning solutions to partial differential equations of the McKean-Vlasov type, and presents a theoretical analysis of the proposed method. In particular, for a certain class of equations, it is shown that the error bound does not increase exponentially in time. This is a significant result that takes advantage of the characteristics of the equations.

**Strengths:**

The uniform bound presented by the authors is a very strong theoretical result. Although I do not understand the details of the proofs, the proofs seem to be reliable because several references are provided. In my opinion, this paper is an excellent paper as a paper in the research area of numerical analysis.

**Weaknesses:**

The most important weakness is that it is not clear what practical applications the McKean-Vlasov type of partial differential equations have.  Unlike numerical analysis, which is a research area in mathematics, there are many people who are interested in applications in the machine learning community. So these results may not be of much interest to them.

Another significant weakness is that the adjoint method and the Monte Carlo method are required in the proposed approach, which are computationally expensive. Considering the computational cost, the proposed method is not considered superior to classical numerical methods, such as the finite difference method.

In addition, the uniform bound seems to hold only for a class of equations, but this is not clearly written in Introduction.

**Questions:**

(1) What are the practical applications of the McKean-Vlasov-type partial differential equations?
(2) Because computationally expensive methods such as the adjoint method and the Monte Carlo method are required, the computational cost of the proposed method may be larger than the classical approaches such as the finite difference method. So, what are the advantages of the proposed neural-network solver compared to the classical numerical methods (e.g, the finite difference method)?

**Limitations:**

No potential negative societal impact is expected.

---

> ### Author Rebuttal · Authors · 2023-08-07
>
> We thank the reviewer for pointing out the need to highlight the application of our research and to compare with classical method. In the following we address the concerns one by one. Our response will be incorporated in our revision.
>
> > What are the practical applications of the McKean-Vlasov-type partial differential equations?
>
> McKean-Vlasov equation describes the evolution of the law of a large interacting particle system. It plays a crucial role in various scientific fields, including physics, control, and social sciences. Just to list a few example applications and how EINN can be utilized accordingly:
>
> - [physics] In the research of plasma physics, MVE with column interaction can be used to model the dynamics of charged electrons in the over-damped limit. In fluid dynamics, the 2D-NSE is a simplification of the 3D Navier-Stokes equation, which is one of the Millennium Prize problems.
> - [control] The mean-field game is formulated as a system of the Hamilton–Jacobi–Bellman equation and the McKean-Vlasov equation. Solving MVE accurately is a crucial part of this research direction. When the interaction between agents is bounded, EINN can directly be used to derive error estimation for solving the MVE part. Please see Theorem 6 in the appendix.
> - [social science] The flocking model was originally designed to model the flocking behavior of birds but now finds application in social sciences, such as in the study of opinion dynamics. When the flocking interaction is bounded, applying the EINN method directly leads to solutions with error estimations.
>
> > What are the advantages of the proposed neural-network solver compared to the classical numerical methods (e.g, the finite difference method)?
>
> In the following, we list some key advantages of EINN compared to the classical methods. Since there are various existing methods, we specifically focus on the finite difference method as pointed out by the reviewer.
>
> - One of the major advantages of EINN and in general NN based PDE solvers is that they do not suffer from the curse of dimensionality (at least not explicitly). Specifically, the finite different method requires to discritize the space so that the solution to the PDE can be represented using a finite dimensional vector. Consequently, the number of grid points grows exponentially with the ambient dimension. For NN based PDE solver, the solution is represented via an NN, which enjoys a significantly stronger representation capability. Mathematically speaking, to approximate a function with sufficiently good regularity, NN is a universal function approximator with width and depth polynomially grows with the ambient dimension. Please also see Appendix A of our paper where we discussed the relation of the classical methods with NN based approaches.
>
> - Further, to maintain the same level of granularity, the number of grid points also grows quickly with the size of the domain. Consequently, if the problem is defined in an unbounded domain, the implementation of the finite difference method requires domain cut-off to ensure the boundedness. In contrast, EINN can directly handle the unbounded cases.
>
> - Another important advantage of EINN over the finite difference method is that, once trained, EINN allows one to access value and gradient of the solution at any spatial coordinate while the finite difference method only provide information on the grid points.
>
> - Recall that at any given time stamp, the solution to MVE is a probability distribution. With EINN, one can sample from the solution at any given time, and our KL-type error estimation ensures that this sampling is meaningful. In contrast, even when the positivity and the preservation of mass is ensured in the finite difference method, it would not generate sample beyond the grid points.
>
> > In addition, the uniform bound seems to hold only for a class of equations, but this is not clearly written in Introduction.
>
> We are not sure what the reviewer means by "this is not clearly written in Introduction", could you be more specific? If the reviewer is referring to the limitations on the interaction kernel, we have clearly stated in Theorem 1 of the introduction that our theoretical guarantee holds for bounded interaction kernel or the singular interaction kernel in the Coulomb or Biot-Savart cases (please see lines 82-83).

---

> > ### Comment · Reviewer_Jhg5 · 2023-08-18
> >
> > Thank you so much for answering my questions. Although some of my concerns have been addressed, it does not seem to me that the benefit explained by the authors is worth the computational cost. My rating is unchanged.

---

> > > ### Author Response · Authors · 2023-08-18
> > > **Response to Reviewer Jhg5 regarding the computational cost**
> > >
> > > We thank the reviewer for expressing again his concern on the computational cost. We agree that this is an important factor and would like to further clarify on this point. The following discussion will be included in our revision.
> > >
> > > Let's break down the computational costs of both EINN and the finite-difference (FD) method for comparison. For the simplicity of the analysis, we assume both EINN and FD are implemented **in a time-continuous fashion, i.e. we directly use their ODE formulation and solve the corresponding ODEs via numerical integrator**. In this way, given the same total evolving time, we can take the computational cost of a single numerical ODE step as the proxy of the total computational cost. Note that both approaches can be implemented in a discrete-time manner, but that would introduce an additional factor due to the time-discretization. We choose to discuss the time-continuous case for the sake of simplicity.
> > >
> > > Notations: Suppose that the ambient space of the PDE is $d$-dimensional. For FD, the computational domain of interest is a $d$-dimensional box, i.e. $[-L, L]^d$; the granularity of FD is $\eta$, i.e. in every dimension there are $L/\eta$ grid points and in total there are $(L/\eta)^d$ grid points. For EINN, let $p$ represent the number of parameters in the neural network, $b$ represent the SGD batch size, $m$ represent the number of particles, and $N$ represent the number of SGD steps.
> > >
> > > For the sake of concreteness, we will take the general MVE
> > >
> > > $$\frac{\partial }{\partial t}\bar \rho(t, x) + \mathrm{div}_x(\bar \rho(t, x) (-\nabla V(x) + K\ast \bar \rho(t, x)) = \Delta \bar \rho(t, x),$$
> > > as example and analyze the overall computational cost of EINN and the finite difference method.
> > >
> > > - [Cost of FD] The solution to FD is represented using the density value over $(L/\epsilon)^d$ grid points. To evaluate the convolution, it takes $O((L/\epsilon)^{2d})$ operations, which dominates the cost of other computations. Consequently, in a single numerical ODE step, the cost of the FD method is $O((L/\epsilon)^{2d}).
> > >
> > > - [Cost of EINN] The cost of evolving the particles is $O(p(b+m))$ since all particles are driven by the hypothesis velocity field, represented using an NN with $p$ parameters and a single forward pass costs $O(p)$ for a single particle. The cost of evolving the score function (see Eq.18 in our paper) is $O(bdp)$ since the every function evaluation of the hypothesis velocity field costs $p$ and we need to perform this over every coordinate in the divergence and over every entry in the SGD mini-batch. The cost for evaluating the interaction is $O(d b m)$, where we recall that every point in the mini-batch of SGD interacts with $m$ particles (see section 2.1). Finally, the cost of computing the gradient using the adjoint method is $O(bdp)$ since the every function evaluation of the hypothesis velocity field costs $p$ and we need to perform this for every coordinate and every entry in the SGD mini-batch (recall that the cost of the back propagation as well as the Jacobian-vector-product of an NN is O(p)). In summary, in a single numerical ODE step in a single SGD step, the cost of EINN is $O(pb + pm + dbm + dbp)$. Additionally, since for EINN we need to run multiple SGD updates, the amortized cost of EINN overall is $O(Npb + Npm + Ndbm + Ndbp)$.
> > >
> > > Based on the above analysis of computational costs, we offer the following discussion:
> > > 1. The cost of finite-difference grows exponentially with the problem dimension $d$. Consequently, EINN has a clear advantage in this setting.
> > > 2. For the low-dimensional case, the comparison between the cost of EINN and finite difference heavily depends on the specific settings of the parameters and there is no unanimous decision on which method is more efficient. Let us just give one concrete example: In our 3D-MVE example, $p \simeq 3000$ (MLP with width 20 and depth 7), $m = 10000$, $b=100$, $N=10000$. So the overall cost of EINN is in the order of $10^{13}$. If we set for finite difference $L=0.1$ (this is a very small domain already), $\eta=1/1000$, we have the overall cost is of the order $10^{13}$. With this example, we do not mean to claim that EINN has the same or even strictly less cost than finite difference, but simply to emphasize finite-difference may not necessarily has a huge advantage over EINN in terms of computational cost even in this low-dimensional setting.
> > >
> > > Thank you again for your time and please let us know if this addresses your concerns.

---

> > > > ### Comment · Reviewer_Jhg5 · 2023-08-20
> > > >
> > > > Thank you very much again for the detailed response. Unfortunately, not all of my concerns have been addressed.
> > > >
> > > > 1. I agree with the authors that the proposed method is perhaps effective for high-dimensional cases; numerical experiments for such high-dimensional problems are preferably provided, though.
> > > >
> > > > 2. The discussion about the computational cost in low-dimensional cases seems to be misleading. I agree with the authors that "the comparison between the cost of EINN and finite difference heavily depends on the specific settings of the parameters and there is no unanimous decision on which method is more efficient." In addition, of course, the application of neural networks to physical simulations is a developing field, and it is not necessary to completely beat classical methods at the present time. However, the specific example provided for the 3D-MVE example seems to be misleading.
> > > > - The authors first estimate the computational complexity for EINN, and then set up the finite difference method so that the computational complexity becomes comparable to that of EINN; however, this gives readers the impression that the finite difference method often requires a similar amount of computation as EINN.
> > > > - The authors estimate the computational cost of the convolution as $O( (L/\varepsilon) ^{2d})$; however, if I understand correctly, this should be $O( d (L/\varepsilon) ^{d}(log (L/\varepsilon) ) )$ because the convolution can be computed by using FFT. So, if $\eta=1/1000$, the computational cost of the finite difference method could be around $10^{7}$ or $10^{8}$.
> > > > - Perhaps, as the authors have noticed, the computational cost is closely related to **the tolerance of numerical errors.** In finite difference methods, at least the second-order method is typically used. If the mesh size is set to $\eta=/1000$, which means the number of nodes in each axis is 100, the error could be around $10^{-4}$, which seems to be more accurate than the results of the proposed method in the paper.  In other words, the finite difference method seems to be able to find more accurate numerical solutions at a rate about, say, $10^5$ times faster. If the tolerance of numerical errors is the same as the numerical errors of the proposed method, the finite difference method is expected to be much faster, even if FFT is not employed for the computation of the convolution.

---

> > > > > ### Author Response · Authors · 2023-08-21
> > > > > **Response  to Reviewer Jhg5 regarding the computational cost**
> > > > >
> > > > > We are glad that the reviewer agrees that in the high-dimensional case, EINN is potentially more efficient than the finite difference method. We also thank the reviewer for pointing out the strategies that to improve the performance of the finite-difference method.
> > > > >
> > > > > > numerical experiments for such high-dimensional problems are preferably provided, though.
> > > > >
> > > > > We will be happy to provide additional numerical results for high-dimensional tasks. However, at short notice, we could not finish the experiment before the author-reviewer discussion deadline. We will be working to provide this result in our revision.
> > > > >
> > > > > > The discussion about the computational cost in low-dimensional cases seems to be misleading.
> > > > >
> > > > > We are sorry that the reviewer find our discussion on the MVE case to be misleading and we acknowledge that there are many strategies to reduce the computational cost of finite difference method. Again, as stated in our previous response, we do not mean to claim that EINN has a computational advantage over the finite-difference methods in low dimensional cases. We simply want to point out that the computational complexity of both approaches are heavily influenced by the choice of parameters and there is no unanimous decision on which method is more efficient, which is acknowledge by the reviewer as well. Moreover, the discussion on whether NN based approach would be a better alternative to the classical method in low dimensional PDE is beyond the scope of our paper (this is basically one of the most important debates in the field). The goal of our paper is mainly theoretical oriented and is to provide rigorous error estimation for MVE with a singular interaction kernel.
> > > > >
> > > > > We thank the reviewer again for providing valuable comments to help us to improve the quality of the paper.

---

### Official Review · Reviewer_HFPv · 2023-07-05

**Soundness:** 3 good
**Presentation:** 3 good
**Contribution:** 3 good
**Rating:** 6
**Confidence:** 4

**Summary:**

The authors propose a novel method for solving the McKean-Vlasov Equation (MVE), a Partial Differential Equation (PDE) that models the dynamics of stochastic particle systems with mean-field interactions. The new approach, named Entropy-dissipation Informed Neural Network (EINN), leverages neural networks to estimate the velocity field underlying the MVE and minimize a proposed potential function. This method is particularly designed to tackle singular interaction kernels, where traditional methods struggle. The authors provide empirical comparisons with state-of-the-art (SOTA) neural network-based MVE solvers and demonstrate the superior performance of the EINN framework. Despite the advancements, a noticeable weakness in this work is the omission of a discussion on the neural operator framework for solving PDEs, which represents a significant progression in the field. This lack of consideration limits the comprehensiveness and depth of the study.

**Strengths:**

1. The paper provides a novel approach to solving MVEs. The idea of using entropy dissipation in the underlying system is inventive and beneficial to the community.
2. The research is solid and thorough. The authors have established theoretical guarantees for their EINN framework and provided clear derivations and proofs.
3. The empirical comparison of EINN with state-of-the-art methods and the positive results strengthen the claim of the proposed method's effectiveness. The authors ensured a fair comparison by using NNs with the same complexity for all methods tested.

**Weaknesses:**

1. The lack of discussion on the neural operator framework for solving PDEs. This approach has been a considerable advancement in the field, and the authors' failure to address or refer to it is an oversight. Further exploration or comparison of the neural operator approach could enhance the comprehensiveness and relevance of the paper.
2. While the authors did a commendable job providing rigorous mathematical proofs, the paper could be difficult for those without a strong background in the relevant mathematical fields. Providing more intuitive explanations or graphical illustrations might have made the methodology more accessible.
3. The empirical testing seems to be limited in scope. More extensive experimentation, including larger-scale problems and more varied examples, would have strengthened the empirical evidence.

**Questions:**

1. While the authors aim to utilize the Lyapunov function to study the deviation of the hypothesis solution from the true solution, they might have overlooked the possibility of overfitting. There seems to be no clear mention of measures taken to avoid overfitting in the model. How does the model ensure that it is generalizable and not just fitting the data it was trained on?
2. How can the EINN framework be adapted or expanded to deal with other types of PDEs?
3. How does the performance of EINN scale with the complexity of the problem, particularly in high dimensional cases?
4. Given the reliance on NNs, how sensitive is the performance of the EINN approach to the choice of NN architecture and parameters?
5. The authors should refer to and discuss the neural operator work for solving PDEs as related work. Here are some references: [Li, Zongyi, et al. "Fourier neural operator for parametric partial differential equations."]; [Kovachki, Nikola, et al. "Neural operator: Learning maps between function spaces."]; [Li, Zongyi, et al. "Neural operator: Graph kernel network for partial differential equations."]; [Gupta, Gaurav, Xiongye Xiao, and Paul Bogdan. "Multiwavelet-based operator learning for differential equations."]; [Xiao, Xiongye, et al. "Coupled Multiwavelet Neural Operator Learning for Coupled Partial Differential Equations."].

**Limitations:**

The authors have specifically designed this approach for solving MVEs, which might limit its general applicability for solving various types of PDEs.

---

> ### Author Rebuttal · Authors · 2023-08-05
>
> We thank the reviewer for pointing out interesting related works. In the following, we address the concerns one by one. The following discussion will be incorporated in our revision.
>
> > The lack of discussion on the neural operator framework for solving PDEs. This approach has been a considerable advancement in the field, and the authors' failure to address or refer to it is an oversight. Further exploration or comparison of the neural operator approach could enhance the comprehensiveness and relevance of the paper.
>
> We thank the reviewer for pointing out this interesting research direction. However, to highlight the major difference between EINN and the approach of the Neural Operator, it's worth noting that they consider **completely different problem settings**: **EINN requires no pre-existing data** and the goal is to obtain the solution to a PDE by solely exploiting the structure of the equation itself. In contrast, **the neural operator approach is data-driven**, i.e. it relies on the existence of configuration-solution pairs. Here, by configuration-solution pairs, we mean the correspondence between some configurations that determine the PDE, e.g. the initial condition or the viscosity parameter in the fluid dynamics problems, and the pre-existing solution to the PDE given the aforementioned configurations. Consequently, the neural operator approach is more like a regression problem where a neural network is trained to learn the abstract map between the configuration and the solution. In contrast, EINN is more like a numerical PDE solver.
>
> Consequently, EINN and the approach of neural operator are two related but quite distinct research directions. They are related in the sense that EINN can provide the data (configuration-solution pairs) required by the neural operator approach. They are distinct since EINN requires no data a priori, while the neural operator approach is built on the supervised learning paradigm.
>
> > While the authors aim to utilize the Lyapunov function to study the deviation of the hypothesis solution from the true solution, they might have overlooked the possibility of overfitting. There seems to be no clear mention of measures taken to avoid overfitting in the model. How does the model ensure that it is generalizable and not just fitting the data it was trained on?
>
> Maybe the reviewer has some misunderstanding of the problem formulation and the proposed approach of our paper.
> As mentioned in the above response, **EINN is not data-driven, i.e. it requires no data** and the goal of our research to solve PDEs by exploiting only its structure. Hence there is no concern of overfitting. According to our theory, a small EINN loss can directly be translated to a high solution accuracy.
>
> > How can the EINN framework be adapted or expanded to deal with other types of PDEs?
>
> We emphasize that this work is mainly theory-oriented and we focus on the special instance of the McKean-Vlasov equation (Eq.2) so that we can establish rigorous error estimations, e.g. Theorems 2 and 3.
>
> In terms of solving PDEs numerically, we emphasize that the EINN framework can be applied to a very broad class of evolutionary PDEs that satisfy the preservation of mass: For any PDE that can be written as a continuity equation (Eq.6) so that the operator $\mathcal{A}$ (Eq.5) is known, one can construct the EINN loss (see Eq.8) accordingly since it only requires the knowledge of $\mathcal{A}$ and $\rho_0$. To solve these PDEs numerically, one simply uses the standard stochastic optimization methods, for example ADAM, to minimize the EINN loss. An important subclass of PDEs that we can solve is the PDEs that admit the structure of Wasserstein gradient flow. This includes the highly general diffusion–advection–interaction equation (see for example the equation below Eq.4.14 in [Santambrogio, F. {Euclidean, metric, and Wasserstein} gradient flows: an overview. Bull. Math. Sci. 7, 87–154 (2017).](https://doi.org/10.1007/s13373-017-0101-1)).
> However, we did not emphasize these numerical examples since we are not able to establish a general error estimation guarantee for all these PDEs, for example our current framework cannot handle the porous medium equation.
>
> > How does the performance of EINN scale with the complexity of the problem, particularly in high dimensional cases?
>
> This is an interesting problem and we intend to test it out in our future work. Currently, this work is mainly theory-oriented and the goal is to establish high quality error estimation for PDEs that involve singular terms.
>
> > Given the reliance on NNs, how sensitive is the performance of the EINN approach to the choice of NN architecture and parameters?
>
> In our experience, EINN is pretty robust to the NN architecture. Some preliminary results (where we vary the width of the NN) are provided in the PDF file. However, for more complicated tasks, e.g. when the solution has worse regularity, we would expect an NN with stronger representation capability would lead to better performance.
>
> > The authors should refer to and discuss the neural operator work for solving PDEs as related work...
>
> We thank the reviewer for pointing out these interesting works and will discuss their relation with EINN in our revision. However, we reemphasize that EINN and the neural operator approach consider completely different settings.

---

> > ### Comment · Reviewer_HFPv · 2023-08-18
> >
> > The authors addressed most of my concerns. I have raised my score to 6.

---

### Official Review · Reviewer_AJKM · 2023-07-12

**Soundness:** 3 good
**Presentation:** 3 good
**Contribution:** 3 good
**Rating:** 7
**Confidence:** 4

**Summary:**

The paper proposed an entropy-dissipation informed neural network for solving certain PDE systems. Using energy dissipation, the paper designed a special structure that propagates information from previous dynamics. They provided a theoretical guarantee for this framework, including the control of the KL divergence between the target and the estimator and a discussion of the approximation error of the NN. They also did some numerical experiments to verify their claims.

**Strengths:**

1. The idea of using energy dissipation to design a framework is very interesting. EINN also takes the previous dynamics precisely and provides a uniform-in-time solution.

2. Mathematically speaking, the proofs are solid and well-written.

3. McKean-Vlasov type PDEs is an important class of PDEs for people from both math and physics communities. The presence of the singular kernel does bring challenges, which are handled by EINN.

**Weaknesses:**

1. The authors specifically take advantage of the special structure of this type of PDEs. On one hand, it performs well for those PDEs; on the other hand, this limits the application of this method to other PDEs.
2. Numerical experiments are done with only comparison with PINN and DRVN.

**Questions:**

1. In Figure 1, first row,  at iteration 0, the value of the objective functions are not the same.
2. In Figure 1, it seems DRVN and EINN perform similarly. Please comment on this.
3. Please comment on how EINN compares with more recent neural-net-based PDE solvers, such as Fourier neural networks.
4. I wonder if more experiments can be done, other than 2D NSE and 3D MVE.

**Limitations:**

yes

---

> ### Author Rebuttal · Authors · 2023-08-05
>
> We thank the reviewer for carefully reading our paper and for making helpful suggestions. In the following, we address the comments/questions one by one. The responses will be incorporated in our revision.
>
> > The authors specifically take advantage of the special structure of this type of PDEs. On one hand, it performs well for those PDEs; on the other hand, this limits the application of this method to other PDEs.
>
> We emphasize that this work is mainly theory-oriented and we focus on the special instance of the McKean-Vlasov equation (Eq.2) so that we can establish rigorous error estimations, e.g. Theorems 2 and 3.
>
> In terms of solving PDEs numerically, we emphasize that the EINN framework can be applied to a very broad class of evolutionary PDEs that satisfy the preservation of mass: For any PDE that can be written as a continuity equation (Eq.6) so that the operator $\mathcal{A}$ (Eq.5) is known, one can construct the EINN loss (see Eq.8) accordingly since it only requires the knowledge of $\mathcal{A}$ and $\rho_0$. To solve these PDEs numerically, one simply uses the standard stochastic optimization methods, for example ADAM, to minimize the EINN loss. An important subclass of PDEs that we can solve is the PDEs that admit the structure of Wasserstein gradient flow. This includes the highly general diffusion–advection–interaction equation (see for example the equation below Eq.4.14 in [Santambrogio, F. {Euclidean, metric, and Wasserstein} gradient flows: an overview. Bull. Math. Sci. 7, 87–154 (2017).](https://doi.org/10.1007/s13373-017-0101-1)).
> However, we did not emphasize these numerical examples since we are not able to establish a general error estimation guarantee for all these PDEs, for example our current framework cannot handle the porous medium equation.
>
> Of course, we acknowledge that our method cannot solve numerically all PDEs. One example is the Poisson equation. However, philosophically, producing highly accurate solutions for PDEs, especially with rigorous error estimation guarantees, is a sophisticated task, which needs to be accomplished by exploiting the underlying structures. Solving all PDEs with a versatile approach like the PINN might not lead to the most accurate solution, as demonstrated in our experiment section.
>
> > In Figure 1, first row, at iteration 0, the value of the objective functions are not the same.
>
> Since the objective functions are different for different methods, it is not surprising that they have a different value initially. In fact, we point out that comparing the function values of different methods may not be meaningful since their objectives are of drastically different meanings.
>
> Instead, we emphasize that the purpose of presenting the function values of different methods is to exclude the possibility that the difference of their solution accuracy (the relative l2 error) is due to insufficient training. Following this logic, to interpret the first column of Figure 1, one should focus on the change of the function value of a single method across different training iterations. Since we observe that all the methods are well trained (the function value sufficiently decreases over a long training period), it is reasonable to infer that the difference of their accuracy is due to the design of the method and the superior solution accuracy of EINN implies that it is a better approach. Please see a similar discussion in lines 352-355 of our paper.
>
> > In Figure 1, it seems DRVN and EINN perform similarly. Please comment on this.
>
> By "perform similarly", maybe the reviewer is referring to the objective value. For this question, please check the discussion above.
>
> However, if the reviewer is referring to the solution accuracy, the experiments (the second and third columns) in Figure 1 show that EINN has a much better solution accuracy compared to DRVN. For example, in the 2D-NSE task, the average relative l2 error of EINN is an order of magnitude smaller than that of DRVN and the final-time-stamp error of EINN is around 1/5 of that in DRVN. Consequently, EINN performs much better.
>
> > Please comment on how EINN compares with more recent neural-net-based PDE solvers, such as Fourier neural networks.
>
> To highlight the major difference between EINN and the approach of Fourier Neural Operator, EINN considers a completely different problem setting as neural operator: **EINN requires no pre-existing data** and the goal is obtain the solution to a PDE by solely exploiting the structure of the equation itself. In contrast, **the neural operator approach is data-driven**, i.e. it relies on the existence of configuration-solution pairs. Here, by configuration-solution pairs, we mean the correspondence between some configurations that determine the PDE, e.g. the initial condition or the viscosity parameter in the fluid dynamics problems, and the pre-existing solution to the PDE given the aforementioned configurations. Consequently, the neural operator approach is more like a regression problem where a neural network is trained to learn the abstract map between the configuration and the solution. In contrast, EINN is more like a numerical PDE solver.
>
> Consequently, EINN and the approach of neural operators are two related but quite distinct research directions. They are related in the sense that EINN can provide the data (configuration-solution pairs) required by the neural operator approach. They are distinct since EINN requires no data a priori, while the neural operator approach is built on the supervised learning paradigm.
>
> > I wonder if more experiments can be done, other than 2D NSE and 3D MVE.
>
> We have conducted an additional experiment on the Porous Medium Equation. While we could not provide rigorous error analysis, the numerical result seems promising. Please see the results in the attached pdf file.

---

> > ### Comment · Reviewer_AJKM · 2023-08-16
> >
> > Thank the authors for addressing all my questions. While some writing could be improved, I do think this work is very novel and interesting. I have raised the rating.

---

### Author Rebuttal · Authors · 2023-08-10

This pdf file contains the additional experiments on the Porous Medium Equation and a preliminary study of the influence of the width of the NN on EINN's performance.

---

### Decision · Program_Chairs · 2023-09-21

**Decision:**

Accept (poster)

**Comment:**

The reviewers unanimously find the paper interesting and result worth publication. The area chair agrees with the evaluation after reading the discussion and the manuscript.